# Factors associated with female infertility in Ethiopia: A systematic review and meta-analysis

**Dagne Addisu** *, **Begizew Yimenu Mekuriaw, Besfat Berihun Erega, Wassie Yazie Ferede, Gedefaye Nibret Mihretie, Enyew Dagnew, Assefa Kebie Mitiku, Tegegne Wale Belachew, Maru Mekie**

Debre Tabor University, Debre Tabor, Ethiopia

* addisudagne7@gmail.com

## Abstract

### Background

Infertility is a significant public health issue that affects couples worldwide. The impacts of infertility is notably higher in Ethiopia due to various factors, such as cultural stigmas surrounding infertility and inadequate infrastructure for diagnosis and treatment. Several fragmented primary studies have assessed factors associated with female infertility in Ethiopia; however, their findings have been controversial and inconclusive. This meta-analysis aimed to identify the factors associated with female infertility in Ethiopia.

### Materials and methods

A comprehensive search was conducted across multiple databases and search engines, including PubMed, African Journals Online, EBSCO, Google Scholar, and the Directory of Open Access Journals. Additionally, studies were searched from the institutional repositories of Ethiopian universities. Data analysis was performed using Stata version 17. The quality of the included studies was evaluated using the Newcastle-Ottawa quality assessment instrument. Heterogeneity and publication bias were assessed using I² and Egger's tests, respectively. A random effects model was employed to identify factors associated with female infertility. The PROSPERO registration number for this meta-analysis was CRD42024525437.

### Result

Six studies were included in the analysis. Factors associated with female infertility included having multiple sexual partners (odds ratio [OR] = 4.31, 95% confidence interval [CI] = 3.46–5.16), a history of sexually transmitted diseases (OR = 2.76, 95% CI = 1.61–3.91), alcohol-abusing partners (OR = 1.57, 95% CI = 1.25–1.89), Khat-abusive partners (OR = 1.85, 95% CI = 1.36–2.35), and women's age over 35 years (OR = 2.07, 95% CI = 1.32–2.81).

**Data availability statement:** All relevant data are within the manuscript and its Supporting Information files.

**Funding:** The author(s) received no specific funding for this work.

**Competing interests:** The authors have declared that no competing interests exist.

## Conclusion

Having multiple sexual partners, a history of sexually transmitted diseases, an alcohol-abusing partner, a khat-abusing partner, and being over the age of 35 were significantly associated with female infertility in Ethiopia. Addressing these risk factors through education, early intervention, lifestyle modifications, and partner involvement can help reduce the burden of infertility and improve the chances of successful conception. The findings underscore the need for further research on understudied factors contributing to female infertility in Ethiopia, including immune function, psychological health, environmental exposures, as well as endocrinological and gynecological conditions.

## Introduction

Infertility defined clinically as inability to become pregnant after one year of unprotected regular sexual intercourse [1,2]. From the perspective of the demographer, infertility is defined as the inability to have a live birth in a woman of reproductive age (15–49 years) with regular unprotected sexual intercourse [3]. It can be classified as either primary or secondary [4]. Primary infertility refers to women who have never conceived, while secondary infertility applies to women who have had at least one successful pregnancy but are unable to conceive again [5].

Globally, infertility affects approximately 60–80 million women [2], and the prevalence rate has increased by 0.37% annually from 1990 to 2017 [6]. A recent meta-analysis reported a global pooled prevalence of infertility at 46.25% [5]. In Africa, the pooled prevalence of primary and secondary infertility was found to be 49.91% and 49.79%, respectively [3]. In Ethiopia, the prevalence of infertility ranges from 24.2% [7] to 27.6% [8].

Various studies have identified several factors associated with infertility, including polycystic ovary syndrome, hormonal imbalances, thyroid disorders, premature ovarian failure, fallopian tube obstructions, congenital malformations, luteal phase defects, endometritis, endometrial polyps, leiomyomas, Asherman's syndrome, chronic systemic diseases, and certain social and personal habits, such as the use of illicit drugs [9,10].

Infertility has significant gynecological, social, and public health consequences [2,9,11]. These consequences can include marital instability, social isolation, sexual abuse, stigma, intimate partner violence, stress, depression, anxiety, and, in some cases, suicide [1,12–14]. In Ethiopia, the impacts of infertility are further worsened by factors such as limited access to healthcare services, high costs of fertility treatments, cultural stigmas surrounding infertility, and inadequate infrastructure for diagnosis and treatment [15]. Women facing infertility may seek alternative solutions, such as visiting spiritualists, traditional healers, or witchcraft practitioners, before resorting to modern medicine [16,17]. Additionally, infertility may lead individuals to engage in extramarital sex as a test of fertility, which exposes them to the risk of sexually transmitted infections [16].

While several primary studies in Ethiopia have investigated factors associated with female infertility, their findings have often been inconsistent and inconclusive [7,15,18–21]. Some studies have reported a strong association between factors like having multiple sexual partner [8] and infertility, while others have not [20]. Similarly, results regarding the impact of having a Khat-abusive partner, an alcohol-abusing partner, and rural residency have been contradictory. These inconsistencies make it difficult to draw definitive conclusions about the factors associated with female infertility in Ethiopia.

Given these gaps in the literature, this study aimed to synthesize available evidence and identify the key factors associated with female infertility in Ethiopia. Specifically, we sought to answer two key research questions: (1) What are the factors associated with female infertility in Ethiopia? (2) How consistent are the relationships between these associated factors and infertility across different studies, and what areas of agreement or divergence exist?

By pooling data from multiple studies, we aimed to provide a comprehensive and reliable understanding of the factors associated with female infertility in Ethiopia. Our findings will help clarify the inconsistencies in the existing literature, highlight specific associated factors that require further investigation, and identify unique contextual factors in Ethiopia that influence infertility. Ultimately, this study may inform targeted interventions and public health strategies to address infertility in the region.

## Materials and methods

### Data reporting and protocol registration

The authors followed the Preferred Reporting Items for Systematic Reviews and Meta-Analyses (PRISMA) checklist to conduct this meta-analysis [22]. The PRISMA checklist is provided as supplementary material in supplementary table 1 (S1 Table). The protocol has been registered in PROSPERO under the registration number CRD42024525437, which can be accessed at https://www.crd.york.ac.uk/PROSPERO.

### Source of information and literature search strategies

The literature search was conducted independently by two authors, DA and BB. Both authors adhered to a standardized protocol that included predefined keywords. The search was carried out across various databases, including PubMed, African Journals Online, EBSCO, Google Scholar, and the Directory of Open Access Journals. Additionally, studies were sourced from the institutional repositories of Ethiopian universities. The reference lists of all included primary studies were also examined to identify any potentially missed studies. To refine their search techniques and locate pertinent studies, the authors employed the Condition, Context, and Population (CoCoPop) framework, with detailed criteria outlined under the inclusion criteria. The searching strategies for PubMed database were (("Infertility, Female"[23] OR "female infertility"[All Fields] OR "infertility in women"[All Fields] OR "women's infertility"[All Fields]) AND ("Risk Factors"[23] OR "determinants"[All Fields] OR "causes"[All Fields] OR "factors"[All Fields] OR "risk factors"[All Fields]) AND ("Ethiopia"[23] OR Ethiopia[All Fields])). The search period spanned from January 1, 2000, to March 20, 2024. Detailed information about the literature search strategies and the search results are provided in Supplementary Table 2 (S2 Table).

### Inclusion criteria

- Condition (Co): Studies that examined the association between different variables and female infertility were included.

- Context (Co): Studies that were conducted in Ethiopia were included.

- Population (Pop): Studies that were done among reproductive age couples were included.

- Study Design: Observational studies (cross-sectional, case-control, and cohort studies) were included.

- Publication Condition: Both published and unpublished studies were included.

- Language: All studies written in English were included.

## Exclusion criteria

- Studies with a different outcome of interest were excluded.

- Studies with irrelevant or unrelated titles were excluded

- Studies with a different study population were excluded.

- Studies that missed important information, such as the adjusted odds ratio (AOR) with a 95% confidence interval for significant variables, were excluded.

- Studies with a low quality assessment score (less than 7 out of 10 for cross-sectional studies and 7 out of 9 for case-control studies, using the Newcastle-Ottawa Quality Assessment Scale) were excluded.

## Study selection and quality assessment process

The data selection process consisted of several key steps. First, all studies obtained from various databases and the institutional repositories of Ethiopian universities were imported into EndNote 7 reference management software. Any duplicate studies were identified and removed using this software.

Two authors, WY and DA, then independently reviewed the remaining studies based on pre-defined inclusion and exclusion criteria. Following this initial review, they convened to discuss their findings, presenting their justifications for including or excluding specific studies while referencing the established criteria. Through this open dialogue, they aimed to reach a consensus regarding the included and excluded studies. In cases where they could not achieve agreement, a third author was consulted to provide an objective perspective. Ultimately, decisions were made collaboratively, ensuring that both authors were aligned on the final list of studies included in the review.

Finally, the quality of the selected studies was assessed independently by WY and DA using the Newcastle-Ottawa Quality Assessment Scale (NOS) [24]. Based on the NOS, articles were considered high quality and included in the review when a study scored at least 7 points out of 10 for cross-sectional studies and 7 out of 9 points for case-control studies. The results of the quality assessment are provided as supplementary data in supplementary table 3 (S3 Table**).**

## Data extraction

After the screening, all pertinent variables were extracted from the included studies by two authors (DA and BB) separately using Microsoft Excel. The authors extracted the name of the first author, the publication status, the year of publication, the study design, the study period, the study region, the sample size, the sampling design, and the adjusted odds ratio (AOR) with a 95% CI for significant factors (Table 1).

## Handling of missing data

Missing data were handled with care to ensure the robustness, accuracy, and replicability of our findings. For studies with missing outcome data or incomplete reporting of relevant statistics (e.g., adjusted odds ratio and 95% confident interval for significant factors), we made efforts to contact the original study authors via email to request the missing information. If we were unable to obtain the missing data after reaching out to the authors, we excluded those specific studies or data points from the meta-analysis, as appropriate. The decision to exclude studies or data points was based on the extent and nature of the missing data; studies with substantial missing data were excluded entirely, while studies with minor missing data were included, provided the missing data did not critically impact the overall analysis.

**Table 1. Shows all data extracted from the primary research for the systematic review and meta-analysis.**

| Study tittle | Date of Data Extraction | Eligibility for Review | Extracted Data |
|---|---|---|---|
| Mekdes et al. Magnitude of infertility and associated factors among women attending selected public hospitals in Addis Ababa, Ethiopia: a cross-sectional study. BMC Women's Health. 2022 2022/01/11;22(1):11. | March 26 to April 6, 2024 | Yes | The name of the first author, Publication status, Publication year, Region, study area, study design, sample size, and AOR with a 95% CI for significant factors (Having multiple sexual partner and alcohol-abusing partners) |
| Rehima et al. Determinants of Primary Infertility Among Married Women Attending Obstetrics and Gynecology Speciality Centers at Adama Town, Oromia, Ethiopia. American Journal of Life Sciences. 2022;10 (1):10–20. | March 26 to April 6, 2024 | Yes | The name of the first author, Publication status, Publication year, Region, study area, study design, sample size, and AOR with a 95% CI for significant factor such Rural residency |
| Zerihun et al. Determinants of Infertility Among Married Women Attending Health Facilities in Bahirdar City, North West Ethiopia 2021 | March 26 to April 6, 2024 | Yes | The name of the first author, Publication status, Publication year, Region, study area, study design, sample size, and AOR with a 95% CI for significant factors (Age greater >35 years and history of STD) |
| Hailegebriel et al. Risk factors of infertility among women attending infertility clinics at st. Paul's hospital millennium medical college in addis ababa 2022. | March 26 to April 6, 2024 | Yes | The name of the first author, Publication status, Publication year, Region, study area, study design, sample size, and AOR with a 95% CI for significant factors(Having Multiple sexual partner, Alcohol-abusing partners, history of STD, alcohol-abusing partners) |
| Desalegn et al. Determinants of Infertility among Married Women Attending Dessie Referral Hospital and Dr. Misganaw Gynecology and Obstetrics Clinic, Dessie, Ethiopia. International Journal of Reproductive Medicine. 2020 2020/03/27;2020: 1540318. | March 26 to April 6, 2024 | Yes | The name of the first author, Publication status, Publication year, Region, study area, study design, sample size, and AOR with a 95% CI for significant factors (Having Multiple sexual partner and history of STD) |
| Nanati et al. The prevalence of infertility and factors associated with infertility in Ethiopia: Analysis of Ethiopian Demographic and Health Survey (EDHS). PLOS ONE. 2023;18(10):e0291912. | March 26 to April 6, 2024 | Yes | The name of the first author, Publication status, Publication year, Region, study area, study design, sample size, and AOR with a 95% CI for significant factors (having high alcohol user partner, rural residency and age > 35 years) |

## Measurement of variables

Alcohol-abusing partners refers to a pattern of alcohol consumption in which an individual regularly consumes more than two standard drinks of alcohol per day [8].

Khat-abusive partner refers to an individual who intentionally ingests khat, a psycho-stimulant substance, at least once per day [18].

## Statistical analysis

Stata 17 was used for the data analysis. The studies' quality was evaluated using the Newcastle-Ottawa quality assessment tool. The assessments of heterogeneity and publication bias were conducted using I-squared statistics and Egger's regression tests, respectively [25,26]. To specify the degree of heterogeneity, we employed the following cut points: I-squared ($I^2$) = 0 indicates no heterogeneity; values of 25%, 50%, and 75% indicate moderate, high, and large heterogeneity, respectively. A random effects model was selected to determine the associated factors of female infertility [27].

## Ethics statement

Since our study was a systematic review and meta-analysis, ethical approval, consent to participate, and consent for publication are not applicable. However, this meta-analysis has been conducted with careful attention to ethical standards and transparency in data handling and analysis. We ensure that all data included are accurate and reliable by critically evaluating each study's methodological quality and identifying any potential biases. Additionally, the results was presented honestly, highlighting both strengths and limitations, and discussing their implications for practice and future research.

## Result

### Search outcome

A total of 57 articles were searched from different databases and search engines (PubMed = 10, google scholar = 4, EBSCO = 5, DOAJ = 1, and AJOL = 37). Additionally, 2 grey literature sources (master's theses) were found in the institutional repositories of Ethiopian universities. Out of the total articles, 8 articles were removed from screening due to duplication. A total of 33 studies were excluded from the review because of irrelevant or unrelated titles and 12 studies were excluded due to different outcomes of interest. Finally, six studies were included to determine the associated factors of female infertility in Ethiopia (Fig 1). The list of all identified articles and the reasons for the exclusion of each study are provided as supplementary data in supplementary Table 4 (S4 Table).

### Characteristics of included studies

A total of six studies were included to identify the factors associated with female infertility in Ethiopia. Among these, four studies were published in different journals, while two studies were grey literature, which are found in Debre Birhan and Bahir Dar University institutional repositories. Regarding study design, three studies were cross-sectional studies, whereas three studies were case-control studies. Regarding the region where the studies were conducted, two studies

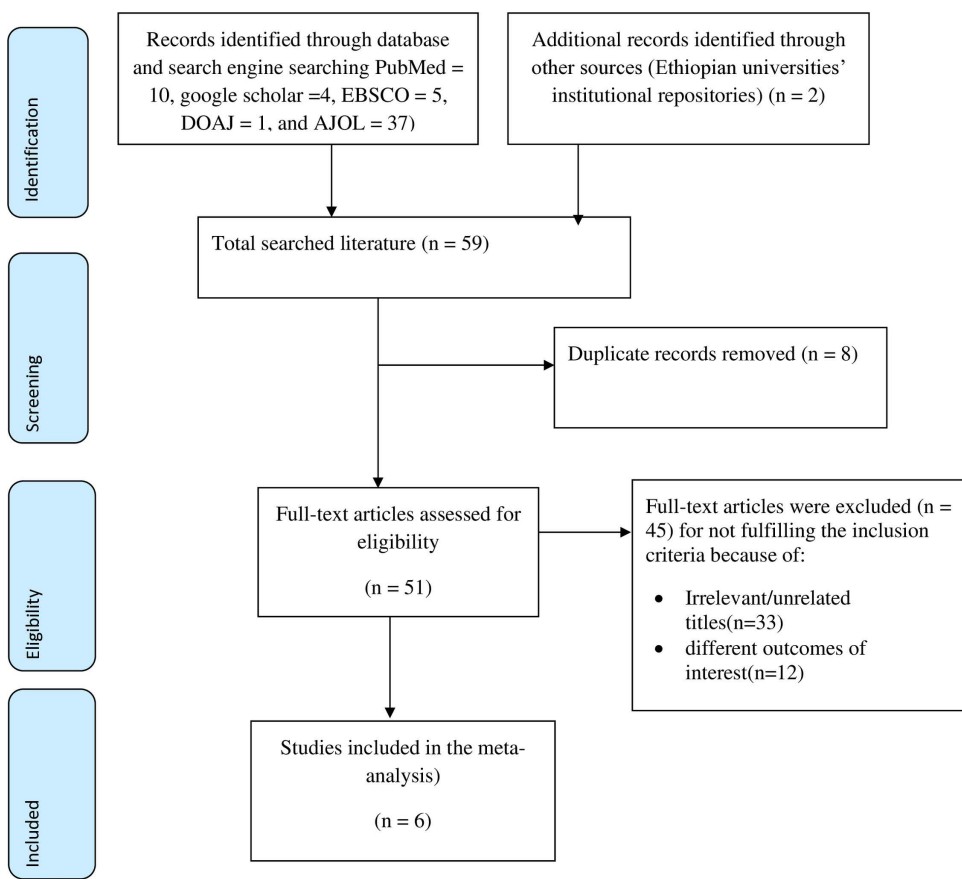

**Fig 1. Schematic representation of the study selection process for identifying primary studies on factors associated with female infertility in Ethiopia.**

were in Addis Ababa, one study from the Oromia region, two studies from the Amhara region, and one study based on EDHS data. A total of 7794 samples were included in this meta-analysis (Table 2).

## Publication bias or small study effect

Egger's regression test was employed to evaluate publication biases, and the results indicated no publication bias or small study effects for having a history of multiple sexual partners (p-value = 0.873), STDs (p-value = 0.918), and having an alcohol-abusive partner (p-value = 0.658). However, only two studies were available to assess the pooled effect of maternal age over 35 years and having a khat-abusive partner, so Egger's regression test was not conducted for these variables.

## Factors associated with female infertility in Ethiopia

This systematic review and meta-analysis focuses on associated factors that were consistently studied and reported across the primary research, with a minimum of two occurrences. While we recognize the significance of immune system function, drug use, lifestyle, diet, psychological stress, socio-economic determinants, hormonal factors (e.g., thyroid disorders, premature ovarian failure, hyperprolactinemia, and polycystic ovary syndrome), and cigarette smoking, these factors were not included in our analysis due to their limited representation in the primary studies.

Six studies reported one or more factors associated with female infertility. The most frequently reported factors were having multiple sexual partners, having alcohol-abusing partner, and having a history of sexually transmitted disease (Table 3).

**Having multiple sexual partner and infertility**: Three studies have examined the association between female infertility and having multiple sexual partners [8,15,19]. A meta-analysis combining these studies found a significant association: women with a history of multiple sexual partners were 4.31 times more likely to experience female infertility compared to those with only one sexual partner (OR = 4.31, 95% CI = 3.46, 5.16). A random-effects model was used to assess this association, revealing high heterogeneity across the studies ($I^2$ = 77.9). The results of this meta-analysis is illustrated in Fig 2.

**The association between having alcohol-abusing partner and female infertility:** three studies found significant association between having alcohol-abusing partner and female infertility [7,8,19]. This meta-analysis also found a significant association between female infertility and having an alcohol-abusing partner. Specifically, women with alcohol-abusing partners were 1.57 times more likely to experience infertility compared to those whose partners did not abuse alcohol (OR = 1.57, 95% CI = 1.25, 1.89). A random-effects model was used to assess the pooled effect, with moderate heterogeneity observed across the studies ($I^2$ = 71.39). This findings are presented in Fig 3.

**Table 2. Characteristics of included studies to assess the factors associated with female infertility in Ethiopia.**

| Author | Publication status | Publication year | Region | study area | study design | sample size |
|---|---|---|---|---|---|---|
| Desalegn et al [15] | Published | 2020 | Amhara | Dessie Referral Hospital and Dr. Misganaw Clinic | case-control | 281 |
| Mekdes et al [8] | Published | 2022 | Addis Ababa | Public hospitals in Addis Ababa | cross-sectional | 441 |
| Rehima et al [18] | Published | 2022 | Oromia | Obstetrics and Gynecology Specialty Centers at Adama Town | case-control | 178 |
| Zerihun et al [20] | Grey literature | N/A | Amhara | Bahir Dar | cross-sectional | 519 |
| Hailegebrel et al [19] | Grey literature | N/A | Addis Ababa | St.paulos Millennium medical college | case-control | 234 |
| Nanati et al [7] | Published | 2023 | EDHS data | Ethiopia | cross-sectional | 6141 |

N/A: Not applicable (grey literature)

**Table 3. Summary of factors associated with female infertility across included studies.**

| Author | Publication year | Factors associated with female infertility | AOR with 95% CI |
|---|---|---|---|
| Mekdes et al [8] | 2022 | Having multiple sexual partner | 3.51[1.64,7.54] |
| | | having alcohol-abusing partner | 1.31[1.11, 1.85] |
| Rehima et al [18] | 2022 | Rural residency | 4[2.07,7.75] |
| Zerihun et al [20] | 2022 | women's age greater than 35 years | 1.69 [1.11, 2.57] |
| | | Having history of STD | 3.63[1.94, 6.8] |
| Hailegebriel et al [19] | 2022 | Having multiple sexual partner | 4.21[3.12, 7.45] |
| | | having alcohol-abusing partner | 2.08[1.24, 3.57] |
| | | Having history of STD | 1.86[1.49, 5.13] |
| | | Having Khat-abusive partner | 2.13[1.65, 4.21] |
| Desalegn et al [15] | 2020 | Having Multiple sexual partner | 5.34[2.12, 13.39] |
| | | Having history of STD | 2.79[1.08, 7.15] |
| Nanati et al [7] | 2023 | having alcohol-abusing partner | 1.55[1.56, 2.28] |
| | | Rural residency | 1.06[1.01, 1.39] |
| | | women's age greater than 35 years | 2.45[1.58, 3.79] |
| | | Having Khat-abusive partner | 1.62[1.12, 2.36] |

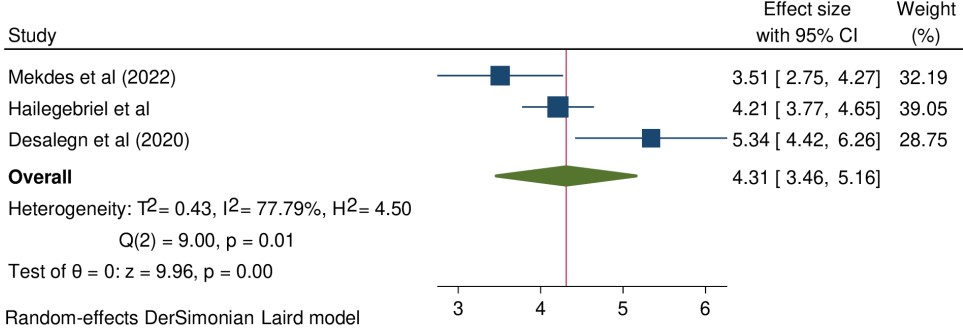

**Fig 2. Forest plot shows pooled odds ratio for the association between female infertility and having multiple sexual partners in Ethiopia.**

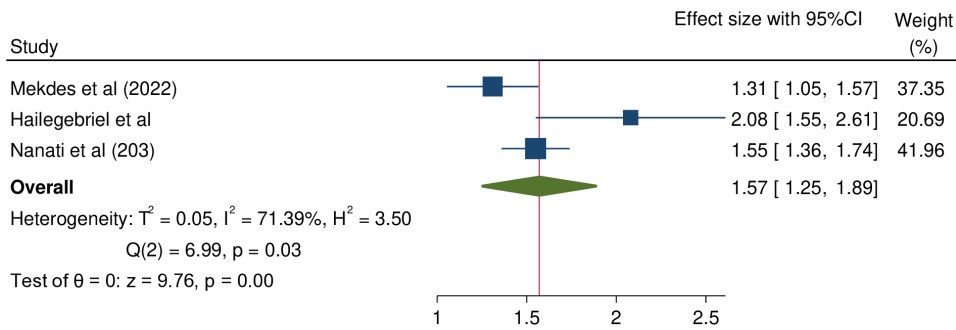

**Fig 3. Forest plot shows pooled odds ratio for the association between female infertility and having alcohol-abusing partner in Ethiopia.**

**Association between history of STD and female infertility:** Three studies have reported the association between a history of STD and female infertility [15,19,20]. Our meta-analysis also found a significant association between infertility and history of STD. Accordingly, those woman who had a history of STD were 2.76 times more likely to have female infertility as compared to their counterpart (OR = 2.76, 95%CI = 1.61, 3.91). High heterogeneity was detected across the studies ($I^2$ = 87.12). The result is shown in Fig 4.

**Association between infertility and women's age greater than 35 years:** Two studies have reported a significant association between women's age over 35 years and infertility [7,20]. This meta-analysis also found a significant association between women's age greater than 35 years and infertility. Accordingly, those women whose age > 35 years were 2.07 times more likely to have infertility (OR = 2.07, 95% CI = 1.32–2.81). High heterogeneity was observed between the studies ($I^2$ = 83.43). The result is shown in Fig 5

**The association between female infertility and having Khat-abusive partner**: A significant association was also noted between female infertility and having khat-abusive partner [7,19]. Those women who had khat abusive partner were 1.85 times more likely to have infertility as compared to their counterpart (OR = 1.85, 95%CI = 1.36, 2.35). Moderate heterogeneity was observed between the studies. The result is shown in Fig 6.

**The association between residency and female infertility:** Two studies have reported a significant association between being rural residency and infertility [7,18]. However, our meta-analysis found no significant association between rural residency and infertility (OR = 2.51, 95%CI = -0.37, 5.39). The result is presented in Fig 7.

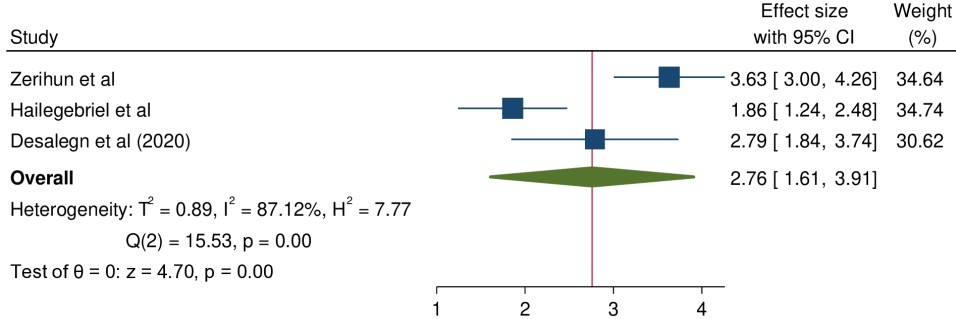

**Fig 4. Forest plot shows pooled odds ratio for the association between history of STD and female infertility in Ethiopia.**

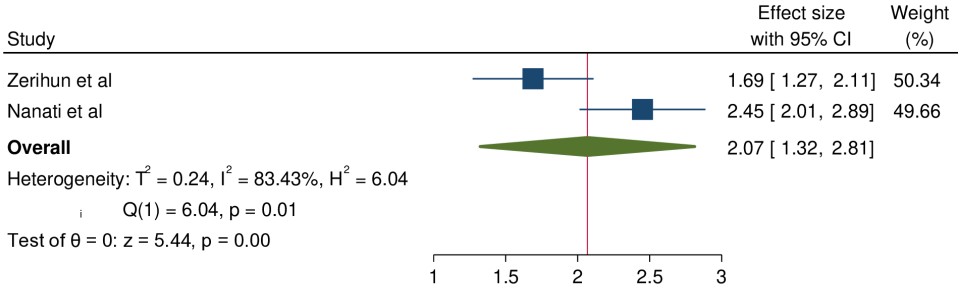

**Fig 5. Forest plot shows the association between women's age over 35 years and infertility.**

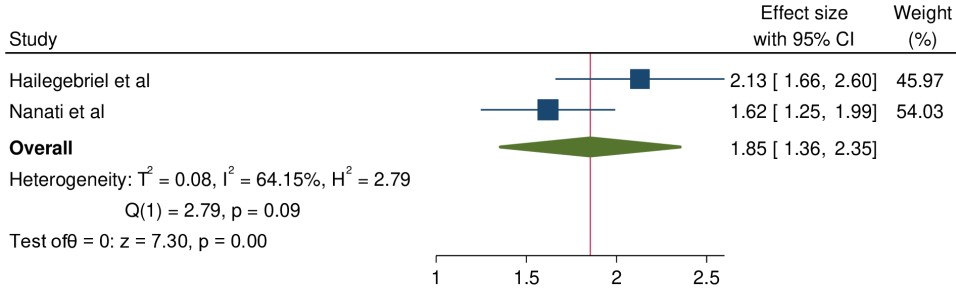

**Fig 6. Forest plot shows the association between female infertility and having khat abusive partners.**

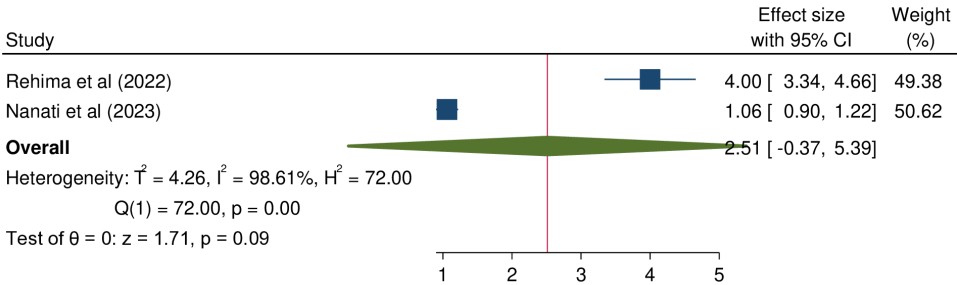

**Fig 7. Forest plot shows the association between being rural residency and female infertility in Ethiopia.**

## Discussion

To the best of our knowledge, this systematic review and meta-analysis is the first of its kind to comprehensively examine the factors associated with infertility in Ethiopia. The findings have the potential to inform both clinical practice and policy development. By identifying key associated factors, this research can help clinicians counsel patients on lifestyle modifications and improve outcomes for women seeking fertility treatment. Furthermore, the results can guide policymakers in developing evidence-based guidelines for the prevention and management of infertility in Ethiopia, ultimately contributing to a reduction in the burden of infertility within the country.

This study identified significant associations between female infertility and several factors, including having multiple sexual partners, a history of sexually transmitted disease, having a partner who abuses alcohol or khat, and being over the age of 35. Conversely, our analysis did not support the significant association of rural residency with infertility, as previously suggested by some studies. This discrepancy highlights the fragmented nature of existing research and emphasizes the importance of our meta-analysis in synthesizing evidence.

Primary studies, along with the results of this meta-analysis, have found a significant association between having multiple sexual partners and infertility. However, it is important to clarify that this is an association, not causation. The number of sexual partners itself does not directly cause infertility. Instead, the increased risk stems from a higher likelihood of exposure to sexually transmitted diseases (STDs), which are known to contribute to fertility problems [28]. Multiple sexual partners can elevate the risk of contracting STDs such as chlamydia, gonorrhea, syphilis, and human papillomavirus. If left untreated, these infections can cause serious reproductive health issues that may impact fertility [28]. Therefore, while having multiple sexual partners is associated with a higher risk of infertility, it is the associated risk of STDs that plays a

significant role. This highlights the importance of practicing safe sex, such as using condoms, and undergoing regular STDs screenings to prevent potential complications that could affect fertility.

A history of sexually transmitted diseases (STDs) was significantly associated female infertility. This finding aligns with a study conducted in Africa, which identified STDs as a major cause of infertility [29]. It is also consistent with an existing systematic review [28] and further supported by another review, which highlights that sexually transmitted infections, particularly chlamydia, are linked to female infertility [30]. STDs such as chlamydia and gonorrhea can lead to pelvic inflammatory disease (PID) if left untreated. PID is an infection of the reproductive organs, including the uterus, fallopian tubes, and ovaries, and can result in scarring and damage. This damage may cause blockages in the fallopian tubes or other structural issues that interfere with the egg's ability to be fertilized or to reach the uterus [28,29]. Chronic inflammation caused by STDs can also disrupt the normal function of the reproductive organs, further impeding fertility [31]. Moreover, infections can affect the cervix, leading to cervical damage that may interfere with sperm transport. These findings highlight the importance of early diagnosis, prevention, and treatment of STDs to reduce infertility risk. Public health strategies should prioritize STD prevention and treatment, particularly in regions with high infection rates, to improve reproductive health outcomes. Additionally, integrating sexual health education and accessible healthcare into reproductive health programs is essential to mitigate the long-term fertility impacts of untreated STDs.

Significant associations have also been noted between infertility and having alcohol-abusing partner. Excessive alcohol consumption can indirectly impact fertility by affecting male fertility, specifically through its negative effects on sperm quality. Studies have shown that alcohol use can lead to a reduction in sperm count, impair sperm motility (the ability of sperm to swim toward the egg), and alter sperm morphology (the shape and structure of sperm). These changes can significantly reduce the chances of successful fertilization. For instance, alcohol-induced oxidative stress can impair spermatogenesis, leading to lower sperm concentration and motility. Additionally, alcohol can cause abnormal sperm morphology, which further diminishes the ability of sperm to fertilize an egg. Thus, a high alcohol consumption partner can contribute to infertility through these multiple pathways related to sperm quality [32,33].

Furthermore, a significant association was observed between women's age greater than 35 years and infertility. This finding contrasts with a previous meta-analysis, which concluded that age was not significantly associated with infertility [5]. However, other studies support the current finding, reinforcing the relationship between advanced women's age and infertility [34,35]. The relationship between advancing women's age and infertility is well-established, with a growing body of research confirming that as women age, their fertility declines due to several biological factors [36,37]. As women approach their mid-30 years and beyond, their fertility begins to decline gradually, primarily due to a decrease in the number and quality of their eggs. Women are born with a finite number of eggs, and this number decreases with age. Moreover, the eggs remaining in the ovaries with advancing age tend to be of lower quality, which can lead to difficulties in conceiving [38]. The decline in egg quality with advancing age is another critical factor [39]. Older eggs have a higher risk of chromosomal abnormalities, which can lead to difficulties in fertilization, poor embryo development, or early pregnancy loss [40,41]. Egg quality is also linked to the likelihood of successful implantation in the uterus, and as a woman's age increases, the chances of implantation success decrease [42,43].

Finally, a significant association was observed between infertility and having Khat-abusive partner. Khat (scientifically known as Catha edulis) is a stimulant plant native to East Africa and the Arabian Peninsula, where it is commonly chewed for its psychoactive effects. The active compounds in khat, particularly cathinone and cathine, have stimulant properties that affect the central nervous system, inducing feelings of euphoria, increased alertness, and heightened energy. However, while khat is culturally ingrained in certain regions, there is growing evidence suggesting that its use can have detrimental effects on male reproductive health, which, in turn, can impact fertility outcomes in couples. Several studies have highlighted the negative effects of khat use on male fertility, particularly its impact on sperm quality and quantity. Chronic khat consumption has been associated with a reduction in sperm count, impaired sperm motility, and abnormal sperm morphology, all of which are critical factors for successful fertilization. Poor semen quality can make it more difficult

for sperm to reach and fertilize the egg, thus increasing the risk of infertility. These effects on sperm parameters may not only reduce the chances of conception but can also contribute to prolonged infertility in couples where the male partner is a regular khat user [23,44,45]. In addition to its impact on sperm quality, chronic khat use has been linked to sexual dysfunction in men [46]. Khat has been shown to interfere with the hormonal regulation of male sexual function, potentially leading to conditions such as erectile dysfunction, reduced libido, and difficulties in achieving or maintaining an erection. These sexual health issues can further impair fertility outcomes, as they may hinder the ability to engage in regular sexual intercourse or affect the timing of conception.

This meta-analysis revealed moderate to high heterogeneity among the included studies, as indicated by the I² statistic reported under each forest plot for the associated factors. This heterogeneity may be attributed to differences in study design, population, and measurement methods across the primary studies. For example, variations in the definition and assessment of associated factors, such as alcohol or khat use could contribute to the observed variability. While random-effects models were used to account for this heterogeneity, the findings should be interpreted with caution.

### Limitation of the study

This study has several limitations that should be considered when interpreting the findings. First, a limited number of studies were included in this review, which may affect the generalizability of the findings. The primary studies included in this meta-analysis also did not assess a comprehensive range of associated factors, such as immune system function [47], drug use [48], lifestyle [49], psychological stress [50], endocrinological conditions (e.g., thyroid disorders [51], premature ovarian failure [52], hyperprolactinemia [53], and polycystic ovary syndrome [52]), uterine fibroids [54], endometrial polyps [55], endometriosis [51], adenomyosis [51] and cigarette smoking [56], all of which could also be associated with female infertility. These limitations highlight the need for future research that includes a broader range of factors, more consistent methodologies, and transparent reporting to provide a clearer and more reliable understanding of the factors associated with female infertility in Ethiopia.

Another limitation of this study is the moderate to high heterogeneity observed among the included studies, which may limit the generalizability of the findings.

Finally, the inability to separate associated factors for primary and secondary infertility in the analysis is another limitation of this study. This lack of separation limits the ability to draw specific conclusions about the unique factors associated with each type of infertility.

### Implication of the findings

The findings of this meta-analysis underscore several critical implications for public health, policy, and healthcare delivery. First, there is a need for public awareness campaigns to address modifiable risk factors, such as multiple sexual partners, unsafe sexual practices, and the impact of alcohol and khat use on reproductive health. Educational initiatives targeting both men and women can promote safer behaviors and healthier lifestyle choices, ultimately reducing the burden of infertility.

From a policy perspective, integrating infertility prevention into broader sexual and reproductive health frameworks is essential. This includes developing clear guidelines for the prevention, screening, and management of sexually transmitted infections (STIs), as well as establishing accessible counseling services for individuals struggling with substance abuse. Given the association between advanced women's age (over 35 years) and infertility, healthcare providers should offer fertility counseling to women, emphasizing the importance of early family planning and the potential risks of delaying childbirth.

To ensure feasibility within Ethiopia's healthcare framework, interventions should be integrated into existing primary healthcare services. This could involve training healthcare workers to deliver reproductive health education and counseling, as well as ensuring that resources for STI management and substance abuse support are accessible in both

urban and rural areas. Strengthening healthcare infrastructure to support reproductive health services and increasing the availability of fertility-related counseling are key components of these efforts. By tailoring interventions to local needs and addressing resource constraints, these measures can be both feasible and effective within the Ethiopian context.

## Conclusion

Having multiple sexual partners, a history of sexually transmitted infections (STIs), an alcohol-abusing partner, a khat-abusing partner, and being over the age of 35 are significantly associated with female infertility in Ethiopia. Encouraging safe sexual practices, regular screening for STIs, avoiding harmful substances like khat, and limiting alcohol consumption can help reduce the burden of infertility and improve the chances of successful conception.

The findings also highlight the need for further research on understudied factors contributing to female infertility in Ethiopia, including immune function, psychological health, environmental exposures, as well as endocrinological and gynecological conditions.

Finally, since the primary studies included in this meta-analysis did not distinguish between factors associated with primary and secondary infertility, future research should differentiate these factors. Such an approach would yield more nuanced insights into the distinct risk factors of each type, ultimately guiding targeted interventions tailored to the specific needs of affected individuals.

## Supporting information

**S1 Table. PRISMA 2009 checklist.**
(DOC)

**S2 Table. Searching strategies for some databases to assess risk factors associated with infertility in Ethiopia.**
(DOCX)

**S3 Table. Quality assessment of articles using Newcastle—Ottawa quality assessment Scale (NOS): (Adapted for cross-sectional and case control studies).**
(DOCX)

**S4 Table. Shows all studies identified in the literature search, including those that were excluded from the analyses.**
(DOCX)

AcknowledgmentWe would like to acknowledge the use of AI models, such as QuillBot and ChatGPT, for their valuable contributions to manuscript editing. The authors have reviewed and verified all content to ensure the accuracy and integrity of the manuscript.

## Author contributions

**Conceptualization:** Dagne Addisu.

**Data curation:** Dagne Addisu, Begizew Yimenu Mekuriaw, Besfat Berihun Erega, Wassie Yazie Ferede, Gedefaye Nibret Mihretie, Enyew Dagnew, Assefa Kebie Mitiku, Tegegne Wale Belachew, Maru mekie.

**Formal analysis:** Dagne Addisu, Begizew Yimenu Mekuriaw, Besfat Berihun Erega, Wassie Yazie Ferede, Gedefaye Nibret Mihretie, Enyew Dagnew, Assefa Kebie Mitiku, Tegegne Wale Belachew, Maru mekie.

**Methodology:** Dagne Addisu, Begizew Yimenu Mekuriaw, Besfat Berihun Erega, Wassie Yazie Ferede, Gedefaye Nibret Mihretie, Assefa Kebie Mitiku, Tegegne Wale Belachew, Maru mekie.

**Project administration:** Dagne Addisu.

**Software:** Begizew Yimenu Mekuriaw, Besfat Berihun Erega, Wassie Yazie Ferede, Gedefaye Nibret Mihretie, Enyew Dagnew, Assefa Kebie Mitiku, Tegegne Wale Belachew, Maru mekie.

**Supervision:** Dagne Addisu.

**Writing – original draft:** Dagne Addisu.

**Writing – review & editing:** Begizew Yimenu Mekuriaw, Besfat Berihun Erega, Wassie Yazie Ferede, Gedefaye Nibret Mihretie, Enyew Dagnew, Assefa Kebie Mitiku, Tegegne Wale Belachew, Maru mekie.

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
