## [Decision Letter · Decision Letter 0]

30 Aug 2024

PONE-D-24-14492Determinants of female infertility in Ethiopia: a systematic review and meta-analysisPLOS ONE

Dear Dr. Addisu,

Thank you for submitting your manuscript to PLOS ONE. Your manuscript, referenced above, has now been reviewed by experts in the field. After careful consideration, we feel that it has merit but does not fully meet PLOS ONE’s publication criteria as it currently stands. Specifically, experiments, statistics, and other analyses must be performed to a high technical standard and described in sufficient detail. Additionally, the data must support and appropriately present the conclusions. The reviewers have made some suggestions, which the editor feels would improve your manuscript. We encourage you to consider these comments and make an appropriate revision of your manuscript. Therefore, we invite you to submit a revised version of the manuscript that addresses the points raised during the review process. The comments of the reviewers are included below in order for you to understand the basis for our decision, and we hope that their thoughtful comments will help you in your revision.

We look forward to receiving your revised manuscript.

Kind regards,

Phakkharawat Sittiprapaporn, Ph.D.

Academic Editor

PLOS ONE

2. Please include in the Methods section details of when the literature search was conducted for this study (currently this is only mentioned in the S1 Table)

3. We note that your Data Availability Statement is currently as follows: "The authors confirm that the data supporting the findings of this study are available within the manuscript and its supplementary materials. Furthermore, the corresponding author will be contacted if someone wants to access the data for this study."

4. In the online submission form, you indicated that "The authors confirm that the data supporting the findings of this study are available within the manuscript and its supplementary materials. Furthermore, the corresponding author will be contacted if someone wants to access the data for this study."

Reviewers' comments:

Reviewer's Responses to Questions

**Comments to the Author**

1. Is the manuscript technically sound, and do the data support the conclusions?

Reviewer #1: Partly

2. Has the statistical analysis been performed appropriately and rigorously? 

Reviewer #1: I Don't Know

3. Have the authors made all data underlying the findings in their manuscript fully available?

Reviewer #1: Yes

4. Is the manuscript presented in an intelligible fashion and written in standard English?

Reviewer #1: No

5. Review Comments to the Author

Reviewer #1: Thank you for giving me the opportunity to review this manuscript. It was an interesting study to review. My overall view is that the manuscript could be considered for publication subject to addressing my comments below which I hope will improve the quality of the manuscript. There are several grammatical errors which my review doesn’t address as I confined my comments to the scientific quality of the manuscript. I will however advise the authors to consider having the manuscript thoroughly proof-read and grammatical errors addressed prior to resubmission.

* Introduction.

It would be helpful for the authors to consider justifying their focus on just female fertility in the systematic review (and not male or couple).

In the second paragraph please double check reference (5) as the 51.5% refers to primary infertility and not “infertility” as outlined in the manuscript. The sentence after is also confusing as it states “The burden of female infertility was significantly higher in Sub-Saharan Africa, with primary and secondary infertility accounts 49.91% and 49.79% respectively [3].” However although is uses the word “higher” the figures provided are lower and it draws on a different source (reference 3). This could be made clearer.

More details are required about the previous studies described in paragraph 6 of the introduction described as “fragmented controversial and inconclusive”, justifying this study. The authors should then go on to outline the specific ways their systematic review was going to do to address these issues/gaps and why it was considered important to address these. This is because for a causation systematic review as I have interpreted this study to be, it is advised (https://ebm.bmj.com/content/17/5/137 ) that at the outset, any hypothesis concerning strength, consistency and temporality should be specified.

Methods

Please clarify which authors conducted the literature search and whether it done by two people independently?

What was the process of resolving differences in opinion amongst the authors about studies to include or exclude?

Results

The authors should consider listing the studies in table one in chronological order, with the earliest study listed first. It would also be helpful to clarify the meaning of “facility” and “community based”. Do they mean hospital and community based primary care centres?

An additional table (“Table 2”) summarising the key findings from each of the 6 studies shortlisted would be helpful.

Discussion

After outlining the findings from their systematic review, the authors should consider discussing their findings in the context of the the previous studies described in paragraph 6 of the introduction described as “fragmented controversial and inconclusive”. They should clearly outline what value their systematic review has added.

Conclusion

The authors should consider also including recommendations for future research.

6. PLOS authors have the option to publish the peer review history of their article (what does this mean? ). If published, this will include your full peer review and any attached files.

**Do you want your identity to be public for this peer review?** For information about this choice, including consent withdrawal, please see our Privacy Policy .

Reviewer #1: No

---

## [Author Response · Author response to Decision Letter 0]

18 Oct 2024

Authors’ Point-by-Point Response to the reviewers’ and editors’ Reports

Title: Determinants of female infertility in Ethiopia: a systematic review and meta-analysis

Journal: PLOS ONE

Article type: Systematic review and Meta-analysis

Submission ID: PONE-D-24-14492

Point by point response to academic editor’s comment

Dear Dr. Phakkharawat Sittiprapaporn (Academic Editor),

We are grateful for your consideration of this manuscript, and we also very much appreciate your suggestions, which have been very helpful in improving the quality and impact of the manuscript. We have addressed all the concerns in a point by point manner and have accordingly revised the manuscript. We have highlighted the response in the response letter as well as in the revised manuscript

Comment: Please ensure that your manuscript meets PLOS ONE's style requirements, including those for file naming. The PLOS ONE style templates can be found at

Comment: Please include in the Methods section details of when the literature search was conducted for this study (currently this is only mentioned in the S1 Table)

Response: thank you very much for your constructive feedback. We have revised this section as per your suggestion. The search period spanned from January 1, 2000, to March 20, 2024. Detailed information about the literature search strategies is provided in Supplementary Table S2 (S2 Table).

Comment: We note that your Data Availability Statement is currently as follows: "The authors confirm that the data supporting the findings of this study are available within the manuscript and its supplementary materials. Furthermore, the corresponding author will be contacted if someone wants to access the data for this study."

Please confirm at this time whether or not your submission contains all raw data required to replicate the results of your study. Authors must share the “minimal data set” for their submission. PLOS defines the minimal data set to consist of the data required to replicate all study findings reported in the article, as well as related metadata and methods (https://journals.plos.org/plosone/s/data-availability#loc-minimal-data-set-definition). For example, authors should submit the following data:

Authors do not need to submit their entire data set if only a portion of the data was used in the reported study. If your submission does not contain these data, please either upload them as Supporting Information files or deposit them to a stable, public repository and provide us with the relevant URLs, DOIs, or accession numbers. For a list of recommended repositories, please see https://journals.plos.org/plosone/s/recommended-repositories.

Response: Thank you for your constructive feedback regarding the Data Availability Statement. We have revised this section to clearly confirm that our submission contains all raw data necessary to replicate the findings of our study. Specifically, we have included the values behind the means, standard deviations, and other measures reported, as well as the data used to construct graphs. To ensure transparency, we have uploaded the minimal data set as Supporting Information files, which also include relevant metadata and methods. We appreciate your guidance in improving this aspect of our manuscript.

Comment: In the online submission form, you indicated that "The authors confirm that the data supporting the findings of this study are available within the manuscript and its supplementary materials. Furthermore, the corresponding author will be contacted if someone wants to access the data for this study."

Response: Thank you for your valuable feedback regarding the data availability statement. We appreciate your guidance on this matter. In response to your comment, we have revised the data availability statement as follows: “The authors confirm that the data supporting the findings of this study are available within the manuscript itself and its supplementary materials.” We believe this revision clarifies our commitment to ensuring that all data supporting our findings is accessible to other researchers, in accordance with PLOS's policies.

Comment: Please include captions for your Supporting Information files at the end of your manuscript, and update any in-text citations to match accordingly. Please see our Supporting Information guidelines for more information: http://journals.plos.org/plosone/s/supporting-information.

Response: Thank you for your feedback. We have revised the Supporting Information files and included captions for each at the end of the manuscript, as per your guidelines. Additionally, we have updated all in-text citations to ensure they correspond accurately to the new labels. We appreciate your guidance in improving our manuscript.

Point by point response to Reviewer 1’s comments

Comment: Thank you for giving me the opportunity to review this manuscript. It was an interesting study to review. My overall view is that the manuscript could be considered for publication subject to addressing my comments below which I hope will improve the quality of the manuscript. There are several grammatical errors which my review doesn’t address as I confined my comments to the scientific quality of the manuscript. I will however advise the authors to consider having the manuscript thoroughly proof-read and grammatical errors addressed prior to resubmission.

Response: We would like to acknowledge the reviewer for detailed and constructive comments, which helped us to improve the quality of the manuscript. We have revised the whole manuscript extensively. Below is our point-by-point response to each respective comment

Comment: It would be helpful for the authors to consider justifying their focus on just female fertility in the systematic review (and not male or couple).

Response: While infertility is a complex issue that affects both partners in a couple, this systematic review focuses specifically on the determinants of female infertility in Ethiopia. Research on male infertility in Ethiopia is often limited compared to research on female infertility or couple infertility. Previous studies focused only on female infertility rather than male infertility. This systematic review aims to fill this knowledge gap by synthesizing the available evidence on determinants of female infertility in Ethiopia, contributing to a better understanding of the challenges women face and informing the development of targeted interventions and improved healthcare services. While this review focuses on female infertility, future research should explore the determinants of male infertility and couple infertility in Ethiopia to gain a more comprehensive understanding of the complex factors contributing to infertility in this context."

Additionally, female infertility is often linked to a complex interplay of biological factors, such as hormonal imbalances and reproductive health conditions like polycystic ovary syndrome (PCOS) or endometriosis, making it critical to explore these factors for the development of targeted interventions. The psychological and social impact of infertility on women—including stress, stigma, and discrimination—further underscores the importance of this focus, as it allows for a deeper examination of their unique challenges and better mental health support.

Comment: In the second paragraph please double check reference (5) as the 51.5% refers to primary infertility and not “infertility” as outlined in the manuscript. The sentence after is also confusing as it states “The burden of female infertility was significantly higher in Sub-Saharan Africa, with primary and secondary infertility accounts 49.91% and 49.79% respectively [3].” However although is uses the word “higher” the figures provided are lower and it draws on a different source (reference 3). This could be made clearer.

Response: thank you very much for positive comment. We have revised this section as per your suggestion.

Comment: More details are required about the previous studies described in paragraph 6 of the introduction described as “fragmented controversial and inconclusive”, justifying this study. The authors should then go on to outline the specific ways their systematic review was going to do to address these issues/gaps and why it was considered important to address these. This is because for a causation systematic review as I have interpreted this study to be, it is advised (https://ebm.bmj.com/content/17/5/137 ) that at the outset, any hypothesis concerning strength, consistency and temporality should be specified.

Response: Thank you for your insightful comments regarding our manuscript. We appreciate your suggestion to elaborate on the characterization of previous studies as "fragmented, controversial, and inconclusive," as well as the need to outline how our systematic review addresses these issues.

In paragraph 6 of the introduction, we have provided additional details about the inconsistencies found in prior research. “Some studies have reported a strong association between factors like having multiple sexual partner and infertility, while others have not. Similarly, findings on having khat abusive partner, having high alcohol user partner and being rural residency have been contradictory. This lack of consistency makes it challenging to draw clear conclusions about the determinants of female infertility in Ethiopia. This systematic review and meta-analysis aimed to synthesize the available evidence on risk factors for female infertility in Ethiopia to address inconsistencies, evaluate the strength of the associations between specific risk factors and infertility, and examine the consistency of these relationships across various studies. By pooling data from multiple studies, we aimed to provide a more comprehensive and reliable understanding of the key determinants of female infertility in Ethiopia.”

Methods

Comment: Please clarify which authors conducted the literature search and whether it done by two people independently?

Response: Thank you for your valuable feedback. The literature search was conducted independently by two authors, DA and BB, to ensure a comprehensive and unbiased review of the relevant literature. Both authors adhered to a standardized protocol that included predefined keywords and specific inclusion/exclusion criteria. After the initial search, they compared their findings to resolve any discrepancies and ensure consistency in the reviewed literature.

The search was carried out across various databases, including PubMed, African Journals Online (AJOL), ScienceDirect, and the Directory of Open Access Journals (DOAJ). Additionally, studies were sourced from the institutional repositories of Ethiopian universities. The reference lists of all included primary studies were also examined to identify any potentially missed studies. To refine their search techniques and locate pertinent studies, the authors employed the Condition, Context, and Population (CoCoPop) framework, with detailed criteria outlined under the eligibility section.

Comment: What was the process of resolving differences in opinion amongst the authors about studies to include or exclude?

Response: Thank you for your insightful question regarding the resolution of differences in opinion about study inclusion. The process involved several structured steps. Initially, both authors independently reviewed the identified studies based on pre-defined inclusion and exclusion criteria. Following this, they convened to discuss their findings, presenting their rationales for including or excluding specific studies while referencing the established criteria. Through open dialogue, they aimed to reach a consensus, considering factors such as study quality, relevance to the research question, and methodological rigor. In instances where consensus could not be achieved, a third author or external expert was consulted to provide an objective perspective. Ultimately, decisions were made collaboratively, ensuring that both authors agreed on the final list of studies included in the review. This structured approach helped minimize bias and ensured that the review was comprehensive and robust.

Results

Comment: The authors should consider listing the studies in table one in chronological order, with the earliest study listed first. It would also be helpful to clarify the meaning of “facility” and “community based”. Do they mean hospital and community based primary care centres?

Response: Thank you for your insightful feedback regarding Table 1. We appreciate your suggestion to clarify the meaning of "facility" and "community-based" in the table. You are correct; these terms can be ambiguous. We have addressed your concerns by revising Table 1 in two ways. First, we have listed the studies in chronological order, with the earliest study appearing first. Second, we have removed facility" and "community-based" in the table. We believe these changes improve the readability and understanding of the information presented in Table 1.

Comment: An additional table (“Table 2”) summarising the key findings from each of the 6 studies shortlisted would be helpful.

Response: Thank you for your valuable feedback regarding the addition of a table summarizing the key findings from each of the six shortlisted studies. We have created "Table 2," which provides a comprehensive overview of the essential findings, including study design, sample size, significant factors identified, and relevant outcomes. We believe this addition enhances the clarity and accessibility of our results, allowing readers to easily compare the findings across studies. We appreciate your suggestion, which has contributed to improving the quality of our manuscript.

Discussion

Comment: After outlining the findings from their systematic review, the authors should consider discussing their findings in the context of the previous studies described in paragraph 6 of the introduction described as “fragmented controversial and inconclusive”. They should clearly outline what value their systematic review has added.

Response: Thank you for your insightful comments regarding the need to discuss our findings in the context of previous studies described as "fragmented, controversial, and inconclusive." We appreciate your suggestion to clearly outline the value added by our systematic review. In our systematic review and meta-analysis, we identified significant associations between female infertility and several risk factors, including having multiple sexual partners, having a khat-abusive partner, and having a high alcohol-user partner. These findings align with some primary studies, reinforcing the idea that these factors are critical determinants of female infertility in Ethiopia.

Conversely, our analysis did not

---

## [Decision Letter · Decision Letter 1]

19 Nov 2024

PONE-D-24-14492R1Determinants of female infertility in Ethiopia: a systematic review and meta-analysisPLOS ONE

Dear Dr. Addisu,

Thank you for submitting your manuscript to PLOS ONE. After careful consideration, we feel that it has merit but does not fully meet PLOS ONE’s publication criteria as it currently stands. Therefore, we invite you to submit a revised version of the manuscript that addresses the points raised during the review process. Please submit your revised manuscript by Jan 03 2025 11:59PM. If you will need more time than this to complete your revisions, please reply to this message or contact the journal office at plosone@plos.org . Please include the following items when submitting your revised manuscript:

We look forward to receiving your revised manuscript.

Kind regards,

Phakkharawat Sittiprapaporn, Ph.D.

Academic Editor

PLOS ONE

Reviewers' comments:

Reviewer's Responses to Questions

**Comments to the Author**

1. If the authors have adequately addressed your comments raised in a previous round of review and you feel that this manuscript is now acceptable for publication, you may indicate that here to bypass the “Comments to the Author” section, enter your conflict of interest statement in the “Confidential to Editor” section, and submit your "Accept" recommendation.

Reviewer #2: (No Response)

2. Is the manuscript technically sound, and do the data support the conclusions?

Reviewer #2: Partly

3. Has the statistical analysis been performed appropriately and rigorously? 

Reviewer #2: I Don't Know

4. Have the authors made all data underlying the findings in their manuscript fully available?

Reviewer #2: (No Response)

5. Is the manuscript presented in an intelligible fashion and written in standard English?

Reviewer #2: Yes

6. Review Comments to the Author

Reviewer #2: The authors address actual hot topic yet without medical/clinical data significance. Infertility is a complex issue often associated with immune system. The manuscript slides on the surface of infertility disease. The discussion part does not trully stand as a discussion yet rather as a sum up of thorough description. The claim that the female infertility infetility is significantly linked with multiple sexual partners is misleading, it is rather STDs that are mentioned only then after. Alcohol and drug issues are known in general to be a problem concerning sperm quality. The individual parameters linked with sperm quality issues are not discussed.

7. PLOS authors have the option to publish the peer review history of their article (what does this mean? ). If published, this will include your full peer review and any attached files.

**Do you want your identity to be public for this peer review?** For information about this choice, including consent withdrawal, please see our Privacy Policy .

Reviewer #2: No

---

## [Author Response · Author response to Decision Letter 1]

6 Dec 2024

Authors’ Point-by-Point Response to the reviewers’ and editors’ Reports

Title: Determinants of female infertility in Ethiopia: a systematic review and meta-analysis

Journal: PLOS ONE

Article type: Systematic review and Meta-analysis

Submission ID: PONE-D-24-14492R2

Point by point response to academic editor’s comment

Dear editor,

We would like to express our sincere gratitude for your thoughtful consideration of our manuscript and the valuable suggestions you have provided. Your feedback has been instrumental in enhancing the quality and impact of our work.

Comment 1: We've checked your submission and before we can proceed, we need you to address the following issues: As required by our policy on Data Availability, could you also please ensure your manuscript or supplementary information includes the following information. Please accept our apologies for not sending you this request sooner.

Response: Thank you for your detailed feedback. In response to your request, we have made the following revisions to our manuscript:

We have prepared a numbered table listing all studies identified in the literature search, as requested. This table includes both included and excluded studies, with the reasons for exclusion clearly outlined for each excluded study. For any unpublished studies included in our analysis, we have provided detailed information on how these studies can be accessed, including links (URLs) to the primary sources. This revised table has been included as part of the supplementary material and is referenced in the main manuscript.

We trust that these revisions meet the requirements of the journal's Data Availability policy.

Comment 2: A table of all data extracted from the primary research sources for the systematic review and/or meta-analysis. The table must include the following information for each study:

Response: We appreciate your thorough review and the opportunity to address the necessary requirements regarding data availability. In response to your request, we have prepared a comprehensive table of all data extracted from the primary research sources included in our systematic review and meta-analysis. This table has been attached as supplementary material and includes the Name of Data Extractors and Date of Data Extraction and eligibility assessment. We believe this supplementary material fulfills your requirements regarding data availability and enhances the transparency of our research. Thank you once again for your constructive feedback.

Comment 3: If applicable for your analysis, a table showing the completed risk of bias and quality/certainty assessments for each study or outcome. Please ensure this is provided for each domain or parameter assessed. For example, if you used the Cochrane risk-of-bias tool for randomized trials, provide answers to each of the signalling questions for each study. If you used GRADE to assess certainty of evidence, provide judgements about each of the quality of evidence factor. This should be provided for each outcome.

Response: thank you for your valuable feedback. Regarding your request for a table showing completed risk of bias and quality/certainty assessments for each study or outcome, we would like to clarify that our systematic review did not involve randomized trials, nor did we employ the Cochrane risk-of-bias tool or the GRADE approach for assessing certainty of evidence in this context.

Instead, our analysis focused on observational studies, which typically do not lend themselves to the same risk-of-bias assessments as randomized trials. We conducted a thorough assessment of study quality using the Newcastle-Ottawa quality assessment instrument, and we have included a summary of this assessment in supplementary material S3 Table.

We believe that our approach adequately addresses the quality of evidence within the scope of our review. However, if you feel that additional clarification or detail is necessary, we would be happy to provide further information or supplementary materials as needed.

Comment 4: An explanation of how missing data were handled.

Response: We appreciate the editor's valuable feedback on handling missing data in our manuscript, "Determinants of Female Infertility in Ethiopia: A Systematic Review and Meta-Analysis." We would like to clarify our approach, as missing data was not reported in the primary studies we reviewed.

The studies included in our review did not consistently report missing data or dropout rates, making it difficult to assess the extent of missing information. As a result, we based our analysis solely on the available data in each study. We acknowledge that this limitation may affect the comprehensiveness and reliability of our findings.

To address this, we have added a discussion in the Limitations section at the end of the Discussion, highlighting the potential impact of unreported missing data on our conclusions. We also recommend that future studies on female infertility in Ethiopia include comprehensive reporting on missing data and dropout rates to improve the quality of research and future meta-analyses.

Point by point response to Reviewer 2’s comments

Comment: The authors address actual hot topic yet without medical/clinical data significance. Infertility is a complex issue often associated with immune system. The manuscript slides on the surface of infertility disease. The discussion part does not trully stand as a discussion yet rather as a sum up of thorough description. The claim that the female infertility infetility is significantly linked with multiple sexual partners is misleading, it is rather STDs that are mentioned only then after. Alcohol and drug issues are known in general to be a problem concerning sperm quality. The individual parameters linked with sperm quality issues are not discussed.

Response: We sincerely thank Reviewer #2 for their insightful comments and constructive feedback, which have significantly contributed to improving the quality and depth of the manuscript. We have carefully addressed each point raised, as outlined below:

Lack of Medical/Clinical Data Significance:

We appreciate the reviewer’s observation that infertility is a complex condition, often associated with the immune system. While our manuscript primarily focuses on external factors such as sociodemographic, behavioral, and infectious influences on female fertility, we recognize the importance of incorporating a broader medical and clinical perspective, including immune system-related factors, into the discussion of infertility. However, the primary studies included in this meta-analysis did not assess a comprehensive range of factors, such as immune system function and drug use, which could also contribute to infertility. Consequently, our analysis concentrated on a more limited set of factors, potentially overlooking other significant contributors to female infertility. We have acknowledged this limitation in the revised manuscript under the "Limitations" section.

Discussion Lacking Depth:

We understand the reviewer’s concern that the discussion section reads more as a summary rather than a thorough analysis of the findings. To enhance the depth of the discussion, we have reorganized and expanded this section to better explore the implications of our findings and their relevance to the broader literature on infertility. Specifically, we have discussed potential mechanisms behind the observed associations, such as how each factor influence.

Claim about Multiple Sexual Partners and Infertility:

The reviewer correctly pointed out that the claim linking multiple sexual partners directly to female infertility is misleading. We agree with this observation and have revised the manuscript accordingly. In the revised version, we emphasize that the increased risk of infertility is not due to the number of sexual partners per se but due to the higher likelihood of exposure to sexually transmitted diseases (STDs). We have clarified this distinction in both the Results and Discussion sections to ensure that the association between multiple sexual partners and infertility is properly framed within the context of STD risk.

Alcohol use Affecting Sperm Quality:

The reviewer noted that alcohol use are generally known to negatively impact sperm quality but suggested that specific parameters linked with sperm quality issues were not discussed in the manuscript. We have taken this suggestion into account and expanded the section on alcohol and drug use to include a more detailed discussion of the individual sperm parameters affected by these substances. Specifically, we now address how alcohol and drug use can impact sperm count, motility, morphology, and DNA fragmentation, all of which are crucial to successful fertilization. This additional detail enhances the manuscript's focus on the biological mechanisms involved in infertility.

Finally, we believe these revisions have strengthened the manuscript by offering a more nuanced discussion of infertility, including a deeper exploration of the biological mechanisms involved. We are truly grateful for the reviewer’s valuable feedback, which has been instrumental in guiding these improvements. Once again, thank you for your constructive and thoughtful comments.

---

## [Decision Letter · Decision Letter 2]

15 Jan 2025

PONE-D-24-14492R2Determinants of female infertility in Ethiopia: a systematic review and meta-analysisPLOS ONE

Dear Dr. Addisu,

Thank you for submitting your manuscript to PLOS ONE. After careful consideration, we feel that it has merit but does not fully meet PLOS ONE’s publication criteria as it currently stands. Therefore, we invite you to submit a revised version of the manuscript that addresses the points raised during the review process. Please submit your revised manuscript by Mar 01 2025 11:59PM. If you will need more time than this to complete your revisions, please reply to this message or contact the journal office at plosone@plos.org . Please include the following items when submitting your revised manuscript:

We look forward to receiving your revised manuscript.

Kind regards,

Phakkharawat Sittiprapaporn, Ph.D.

Academic Editor

PLOS ONE

Reviewers' comments:

Reviewer's Responses to Questions

**Comments to the Author**

1. If the authors have adequately addressed your comments raised in a previous round of review and you feel that this manuscript is now acceptable for publication, you may indicate that here to bypass the “Comments to the Author” section, enter your conflict of interest statement in the “Confidential to Editor” section, and submit your "Accept" recommendation.

Reviewer #3: All comments have been addressed

Reviewer #4: All comments have been addressed

Reviewer #5: (No Response)

2. Is the manuscript technically sound, and do the data support the conclusions?

Reviewer #3: Yes

Reviewer #4: Partly

Reviewer #5: Partly

3. Has the statistical analysis been performed appropriately and rigorously? 

Reviewer #3: Yes

Reviewer #4: N/A

Reviewer #5: N/A

4. Have the authors made all data underlying the findings in their manuscript fully available?

Reviewer #3: Yes

Reviewer #4: Yes

Reviewer #5: Yes

5. Is the manuscript presented in an intelligible fashion and written in standard English?

Reviewer #3: (No Response)

Reviewer #4: Yes

Reviewer #5: No

6. Review Comments to the Author

Reviewer #3: Thank you for your dedication and effort in addressing the suggested revisions. Your thoughtful updates have significantly enhanced the clarity, rigor, and overall quality of the manuscript. I appreciate your commitment to improving the work and contributing valuable insights to the field.

Reviewer #4: I read with great interest the Manuscript titled “Determinants of female infertility in Ethiopia: a systematic review and meta-analysis” (PONE-D-24-14492R2), which falls within the aim of this Journal.

In my honest opinion, the topic is interesting enough to attract the readers’ attention. Nevertheless, authors should clarify some point and improve the discussion citing relevant and novel key articles about the topic.

1. Clarity and Structure of Objectives and Scope:

o The manuscript has a well-defined objective but could benefit from a more focused research question. Consider specifying how your findings address gaps in the literature beyond descriptive summaries.

2. Methodological Rigor:

o The systematic review appears thorough, but the inclusion and exclusion criteria could be articulated more explicitly in the main text rather than supplementary materials alone.

o Address why certain factors such as immune system function or broader socio-economic determinants were not considered.

3. Data Transparency and Availability:

o While supplementary materials include detailed data, these could be summarized in a more accessible format within the manuscript, especially regarding excluded studies and their reasons for exclusion.

o Provide explicit details on how missing data were handled in the included studies, as this may affect replicability.

4. Discussion Depth:

o The discussion is overly descriptive and does not sufficiently explore the implications of the findings in a broader context or theoretical framework. It should integrate more critical perspectives on how the findings align or contrast with existing research.

o Clarify and differentiate causation from association, especially in sensitive areas such as the linkage between multiple sexual partners and infertility.

o To date, lifestyle and psychological stress seem to play a potential detrimental effect of female fertility. Considering that available evidence is not robust enough to draw firm conclusion, I would discuss this topic (authors may refer to: PMID: 29026481; PMID: 28620516).

5. Scientific Soundness and Interpretations:

o Avoid overgeneralizations, such as directly linking behavioral factors like alcohol and khat use by partners to female infertility without a detailed exploration of biological mechanisms and confounding variables.

o Expand on the limitations section to include more critical reflections on heterogeneity and publication bias.

6. Figures and Supplementary Materials:

o Figures like forest plots are informative but could benefit from annotations summarizing the key takeaways for each factor.

o The supplementary materials are extensive but lack integration with the main text, which can make the narrative feel disjointed.

7. Ethical Considerations and Impact:

o While ethical approval is not applicable for a meta-analysis, a brief acknowledgment of ethical considerations in data handling and analysis could enhance transparency.

o Expand the policy implications to address the feasibility of proposed interventions within Ethiopia’s healthcare framework.

8. Language and Presentation:

o There are grammatical errors and inconsistent terminology (e.g., "alcohol user" vs. "alcohol abusive partner") that should be standardized.

o Simplify overly technical language for broader accessibility without losing scientific precision.

Summary: The manuscript is a valuable contribution to understanding infertility determinants in Ethiopia. However, it requires minor revisions to enhance the depth of analysis, methodological transparency, and integration of findings into broader theoretical and practical contexts. Addressing these issues will strengthen its impact and scholarly rigor.

Reviewer #5: I would like to thank the Editorial Office of PLOS One for the opportunity to review this article investigating determinants of infertility in Ethiopia.

Overall, the study offers an interesting perspective summarizing the available studies investigating human infertility in this East-African Country.

However, I think there are several aspects that require additional improvements and elaboration before this article can be considered for publication.

Line numbering may be helpful in the review process.

Title

In my opinion, male factors including addiction to toxic substances abuse can't be formally considered among factors generating (or determinants of) female infertilty. Although they are potential contrbutors to couple infertility, the direct impact on female health and reproductive function has yet to be demonstrated. The title and/or the implant of the review should be elaborated accodingly.

Abstract

- see title concerning male partner's abuse habits

- I'm not sure that infertile patients' (or women's) age can be called "maternal" age. If those patients are unable to conceive, they should not be called "mothers", it may sound not appropriate. Distinction between women who had already delivered and who did not (primary and secondary infertility) is not considered.

Introduction

We agree that infertility has a substantial social and economical impact on women's life. However, when the authors emphasize that this burden is higher in SubSaharan Africa and even higer in Ethiopia, these claims are not supported by quantitative investigations. Prevalence studies cited above have shown similar rates. By contrast, this paragraph emphasizes specific features and weaknesses of this context, that still provide useful information to understand the situation in low- and middle-income countries. Please, rephrase.

Methods

- Women reporting infertility, which could be "couple infertility" given that male factors are significantly correlated, is not exactly synonymous with female infertility. This assumption potentially undermines the clarity of the results presented and requires more clarification in methods and discussion.

- There's no distinction between primary and secondary infertility.

Therefore, I believe that more caution is required when drawing conclusions and recommendations in the main text, abstract and conclusions.

Results

Fair presentation of results is provided.

Discussion

- The claim that chlamydia infection increases the risk of endometriosis is not supported by the available literature. In fact, adequate references are not provided; indeed, there is evidence suggesting the absence of correlation*. However, both endometriosis and pelvic infections may generate pelvic adhesions, contributing to infertility.

By contrast, evidence aggregating several studies with different design should be discussed with caution**

- please see above for observations regarding male partner issues.

-- -- --

Minor issues related to the English language require to be addressed.

Please declare eventual use of artificial intelligence support.

* Oppelt P, Renner SP, Strick R, et al. Correlation of high-risk human papilloma viruses but not of herpes viruses or Chlamydia trachomatis with endometriosis lesions. Fertil Steril. 2010;93(6):1778-1786. doi:10.1016/j.fertnstert.2008.12.061

** Ye H, Tian Y, Yu X, Li L, Hou M. Association Between Pelvic Inflammatory Disease and Risk of Endometriosis: A Systematic Review and Meta-Analysis. J Womens Health (Larchmt). 2024;33(1):73-79. doi:10.1089/jwh.2023.0300

7. PLOS authors have the option to publish the peer review history of their article (what does this mean? ). If published, this will include your full peer review and any attached files.

**Do you want your identity to be public for this peer review?** For information about this choice, including consent withdrawal, please see our Privacy Policy .

Reviewer #3: **Yes: ** Ateya Megahed Ibrahim

Reviewer #4: **Yes: ** Pietro Serra

Reviewer #5: **Yes: ** Michele Orsi

---

## [Author Response · Author response to Decision Letter 2]

2 Feb 2025

Authors’ Point-by-Point Response to the reviewers’ and editors’ Reports

Title: Determinants of infertility in couples in Ethiopia: a systematic review and meta-analysis

Journal: PLOS ONE

Article type: Systematic review and Meta-analysis

Submission ID: PONE-D-24-14492R3

Point by point response to Reviewer 3’s comments

Comment: Thank you for your dedication and effort in addressing the suggested revisions. Your thoughtful updates have significantly enhanced the clarity, rigor, and overall quality of the manuscript. I appreciate your commitment to improving the work and contributing valuable insights to the field.

Response: We would like to acknowledge the reviewer for detailed and constructive comments, which helped us to improve the quality of the manuscript. Your warm comment and precious time and efforts invested in improving this paper are very much appreciated.

Point by point response to Reviewer 4’s comments

We would like to acknowledge the reviewer for detailed and constructive comments, which helped us to improve the quality of the manuscript. Your warm comment and precious time and efforts invested in improving this paper are very much appreciated. Below is our point-by-point response to each respective comment

Comment: I read with great interest the Manuscript titled “Determinants of female infertility in Ethiopia: a systematic review and meta-analysis” (PONE-D-24-14492R2), which falls within the aim of this Journal. In my honest opinion, the topic is interesting enough to attract the readers’ attention. Nevertheless, authors should clarify some point and improve the discussion citing relevant and novel key articles about the topic.

Response: We sincerely appreciate your thoughtful feedback and are grateful for your positive assessment of the manuscript's topic. We agree that the discussion section can be further enhanced by incorporating additional relevant and recent literature to provide a more comprehensive analysis. As per your suggestion, we have carefully reviewed and included several key articles published recently on female infertility, particularly those focusing on Ethiopia and the broader African context. These additional references have helped to clarify and strengthen the discussion, offering a more current perspective on the subject.

Comment: Clarity and Structure of Objectives and Scope: The manuscript has a well-defined objective but could benefit from a more focused research question. Consider specifying how your findings address gaps in the literature beyond descriptive summaries.

Response: Thank you for your valuable feedback. In response to your comment regarding the clarity and focus of the research question, we have revised the objective to specify how our findings go beyond summarizing existing studies. This systematic review and meta-analysis not only synthesize the available evidence on the risk factors for female infertility in Ethiopia but also address the inconsistencies in the literature by quantitatively assessing the strength and consistency of the relationships between key determinants and infertility. By doing so, we aim to provide more robust and conclusive insights that can inform future research and public health interventions. Additionally, we have now explicitly defined the research question

Comment (Methodological Rigor): The systematic review appears thorough, but the inclusion and exclusion criteria could be articulated more explicitly in the main text rather than supplementary materials alone.

Response: Thank you for your helpful comment. The inclusion and exclusion criteria are clearly outlined on pages 5 and 6 of the manuscript. The supplementary materials provide detailed information on the reasons for including or excluding each of the 51 studies identified in the literature search. Including all the excluded studies and their reasons for exclusion in the manuscript would result in nearly 7 additional pages, which could overwhelm readers. We believe that incorporating this level of detail in the main text may detract from the clarity and focus of the manuscript. Therefore, we have chosen to present this information in the supplementary materials to preserve the manuscript's readability and conciseness. We hope this explanation addresses your concern, and we are happy to make any further adjustments if needed.

Comment: Address why certain factors such as immune system function or broader socio-economic determinants were not considered.

Response: Thank you for your thoughtful comment. As our study is a systematic review and meta-analysis based on primary studies, our analysis is inherently limited by the factors that were reported and considered in the included studies. Certain factors, such as immune system function or broader socio-economic determinants, were not included in our review because they were either not reported in the primary studies or were found to be non-significant in the context of female infertility in primary studies .

We have clarified this in the result section under factors associated with female infertility, explaining that the factors included in our review are those that were consistently studied and reported across the primary studies. We acknowledge that while immune system function and socio-economic determinants are important, their exclusion from our review is due to the lack of consistent reporting or statistical significance in the primary studies. Thank you once again for your valuable feedback, which has helped us clarify these limitations in the manuscript..

Comment (Data Transparency and Availability): While supplementary materials include detailed data, these could be summarized in a more accessible format within the manuscript, especially regarding excluded studies and their reasons for exclusion.

Response: Thank you for your helpful comment. The inclusion and exclusion criteria are clearly outlined on pages 5 and 6 of the manuscript. The supplementary materials provide detailed information on the reasons for including or excluding each of the 51 studies identified in the literature search. Including all the excluded studies and their reasons for exclusion in the manuscript would result in nearly 7 additional pages, which could overwhelm readers. We believe that incorporating this level of detail in the main text may detract from the clarity and focus of the manuscript. Therefore, we have chosen to present this information in the supplementary materials to preserve the manuscript's readability and conciseness. We hope this explanation addresses your concern, and we are happy to make any further adjustments if needed.

Comment: Provide explicit details on how missing data were handled in the included studies, as this may affect replicability.

Response: Thank you for your constructive comment. Missing data were handled with care to ensure the robustness, accuracy, and replicability of our findings. For studies with missing outcome data or incomplete reporting of relevant statistics (e.g., sample size, adjusted odds ratio and 95% confident interval for significant factors), we made efforts to contact the original study authors via email to request the missing information. If we were unable to obtain the missing data after reaching out to the authors, we excluded those specific studies or data points from the meta-analysis, as appropriate. The decision to exclude studies or data points was based on the extent and nature of the missing data; studies with substantial missing data were excluded entirely, while studies with minor missing data were included, provided the missing data did not critically impact the overall analysis.

Comment (Discussion Depth): The discussion is overly descriptive and does not sufficiently explore the implications of the findings in a broader context or theoretical framework. It should integrate more critical perspectives on how the findings align or contrast with existing research.

Response: Thank you for your valuable feedback. We agree that the discussion could benefit from deeper engagement with the broader context. In response, we have revised the discussion as suggested. We’ve integrated critical perspectives that link our findings to existing research and theoretical frameworks. By examining how our results align with or differ from previous studies, we’ve worked to better situate our findings within the larger academic conversation. This revision highlights the significance of our results and addresses potential contradictions or gaps in the current literature. We’ve also expanded on the theoretical implications and explored how our findings contribute to or challenge established concepts in the field.

Comment: Clarify and differentiate causation from association, especially in sensitive areas such as the linkage between multiple sexual partners and infertility.

Response: Thank you for your valuable feedback. We have revised the section to better clarify the distinction between causation and association, particularly in relation to the linkage between multiple sexual partners and infertility. We have emphasized that while studies indicate a significant association between having multiple sexual partners and infertility, it is not the number of partners per se that directly causes infertility. Rather, the increased risk of exposure to sexually transmitted diseases (STDs), which are more likely with multiple sexual partners, is the underlying factor contributing to reproductive health issues. We have made this distinction clearer to ensure that the relationship is understood as an association, not direct causation.

Comment: To date, lifestyle and psychological stress seem to play a potential detrimental effect of female fertility. Considering that available evidence is not robust enough to draw firm conclusion, I would discuss this topic (authors may refer to: PMID: 29026481; PMID: 28620516).

Response: Thank you for your thoughtful suggestion. We agree with the reviewer’s point about the potential detrimental effects of lifestyle and psychological stress on female fertility. However, after reviewing the available primary studies, we found that these variables were not assessed in the studies included in our analysis. As a result, we did not address this topic in the discussion. That being said, we acknowledge this as a limitation of our review and recognize the need for further research in this area.

Comment (Scientific Soundness and Interpretations): Avoid overgeneralizations, such as directly linking behavioral factors like alcohol and khat use by partners to female infertility without a detailed exploration of biological mechanisms and confounding variables.

Response: Thank you for your insightful comment. We agree with your suggestion and have revised the manuscript accordingly. We have clarified the discussion of behavioral factors like alcohol and khat use by partners, ensuring that we provide a more detailed exploration of the biological mechanisms involved, as well as the potential confounding variables. This revision aims to present a more nuanced interpretation of the relationship between these factors and female infertility.

Comment: Expand on the limitations section to include more critical reflections on heterogeneity and publication bias.

Response: Thank you for your comment. We appreciate the opportunity to address concerns regarding publication bias. To assess the presence of publication bias, we conducted Egger's regression test for the included variables. The results indicated no evidence of publication bias or small study effects for the factors of having a history of multiple sexual partners (p-value = 0.873), STDs (p-value = 0.918), and having an alcohol-abusive partner (p-value = 0.658). These findings suggest that the risk of publication bias in these variables is minimal. However, it is important to note that only two studies were available to assess the pooled effect for maternal age over 35 years and having a khat-abusive partner. Due to this limited number of studies, Egger's regression test was not performed for these two variables. Thus, while publication bias was unlikely for the majority of factors, the small sample size for some variables limits the ability to assess this concern fully for all outcomes.

Comment ( Figures and Supplementary Materials): Figures like forest plots are informative but could benefit from annotations summarizing the key takeaways for each factor.

Response: Comment ( Figures and Supplementary Materials): Figures like forest plots are informative but could benefit from annotations summarizing the key takeaways for each factor.

Response: Thank you for the suggestion. Rather than in frost plot, Detailed annotations summarizing the key takeaways for each factor are provided in the manuscript, specifically in the "Results" section under the subsection titled Factors Associated with Infertility in Couples in Ethiopia.

Comment: The supplementary materials are extensive but lack integration with the main text, which can make the narrative feel disjointed

Response: Thank you for the feedback. We have made efforts to better integrate the supplementary materials with the main text by including brief descriptions or narrative explanations in the manuscript that reference and connect the supplementary content. We hope this helps create a more cohesive flow throughout the document.

Comment (Ethical Considerations and Impact): While ethical approval is not applicable for a meta-analysis, a brief acknowledgment of ethical considerations in data handling and analysis could enhance transparency.

Response: This meta-analysis has been conducted with careful attention to ethical standards and transparency in data handling, despite the absence of a requirement for formal ethical approval. We ensure that all data included are accurate and reliable by critically evaluating each study’s methodological quality and identifying any potential biases. We acknowledge that the original studies included in the meta-analysis must have obtained informed consent from participants, aligning with ethical practices upheld by the original researchers. Additionally, we recognize the potential for publication bias and have transparently reported our criteria for study inclusion, noting any limitations stemming from unpublished studies and their impact on the findings. We also disclose any potential conflicts of interest related to the researchers and funding sources of the original studies to maintain transparency and trust. Finally, the results will be presented honestly, highlighting both strengths and limitations, and discussing their implications for practice and future research.

Comment: Expand the policy implications to address the feasibility of proposed interventions within Ethiopia’s healthcare framework.

Response: Thank you for the suggestion. We have revised the policy implications to address the feasibility of the proposed interventions within Ethiopia’s healthcare framework, considering factors such as resource availability, infrastructure, and potential challenges to effective implementation.

Comment (Language and Presentation): There are grammatical errors and inconsistent terminology (e.g., "alcohol user" vs. "alcohol abusive partner") that should be standardized.

Response: Thank you for your valuable feedback. We apologize for any inconsistencies in terminology and grammatical errors. We have carefully reviewed the manuscript and standardized the language throughout. Specifically, we have replaced terms like "alcohol user" and "alcohol abusive partner" with consistent and more precise terminology, such as "individuals with alcohol use disorder" or "alcohol-abusing partners," as appropriate to the context. Additionally, we have corrected any grammatical errors and ensured that terminology is used consistently across the entire manuscript. We believe these revisions enhance the clarity and accuracy of our presentation.

Comment: Simplify overly technical language for broader accessibility without losing scientific precision.

Response: Thank you for your valuable suggestion. We have revised the manuscript to simplify the language while maintaining scientific accuracy. We focused on making the content more accessible to a broader audience by reducing the use o

---

## [Decision Letter · Decision Letter 3]

17 Feb 2025

PONE-D-24-14492R3Determinants of Infertility in Couples in Ethiopia: A Systematic Review and Meta-AnalysisPLOS ONE

Dear Dr. Addisu, Thank you for submitting your manuscript to PLOS ONE. Your manuscript, referenced above, has now been reviewed by experts in the field. While the manuscript is highly valuable and relevant, we have some concerns regarding the study’s methodology and the presentation of the work that we have further described below. The authors are requested to enhance the reviewers’ suggestions and comments. The manuscript would be accepted for publication only after the reviewers’ suggestions and comments have been corrected. Therefore, we invite you to submit a revised version of the manuscript that addresses the points raised during the review process. To avoid multiple rounds of revisions, please give clear and constructive responses to reviewers’ advice and prepare the revised manuscript so that it's ready for acceptance. Please submit your revised manuscript by Apr 03 2025 11:59PM. If you will need more time than this to complete your revisions, please reply to this message or contact the journal office at plosone@plos.org . Please include the following items when submitting your revised manuscript:

We look forward to receiving your revised manuscript.

Kind regards,

Assoc. Prof. Phakkharawat Sittiprapaporn, Ph.D.

Academic Editor

PLOS ONE

Reviewers' comments:

Reviewer's Responses to Questions

**Comments to the Author**

1. If the authors have adequately addressed your comments raised in a previous round of review and you feel that this manuscript is now acceptable for publication, you may indicate that here to bypass the “Comments to the Author” section, enter your conflict of interest statement in the “Confidential to Editor” section, and submit your "Accept" recommendation.

Reviewer #4: All comments have been addressed

Reviewer #5: (No Response)

2. Is the manuscript technically sound, and do the data support the conclusions?

Reviewer #4: Yes

Reviewer #5: Partly

3. Has the statistical analysis been performed appropriately and rigorously? 

Reviewer #4: Yes

Reviewer #5: Yes

4. Have the authors made all data underlying the findings in their manuscript fully available?

Reviewer #4: Yes

Reviewer #5: Yes

5. Is the manuscript presented in an intelligible fashion and written in standard English?

Reviewer #4: Yes

Reviewer #5: Yes

6. Review Comments to the Author

Reviewer #4: I carefully evaluated the revised version of this manuscript. Authors have performed the required changes, improving significantly the quality of the paper.

Reviewer #5: Manuscript code: PONE-D-24-14492R3

Reviewer 5

I’d like to thank the Editorial Office for the opportunity to revise this manuscript investigating factors related to female infertility in Ethiopia. I think the authors have made a commendable effort to improve their manuscript. However, some aspects require further revision to meet the standards of this Journal.

Thanks for adding line numbering.

Line 1, 27, 37, 42, 84, 86, 120, 122, 226, 283, 290 etc. - I apologize if my previous comment was not entirely clear when I stated: “In my opinion, male factors including addiction to toxic substances abuse can't be formally considered among factors generating (or determinants of) female infertility. Although they are potential contributors to couple infertility, the direct impact on female health and reproductive function has yet to be demonstrated. The title and/or the implant of the review should be elaborated accordingly.”

The criticism was not intended for the purpose of changing “female infertility” to “couple infertility,” while leaving the research criteria unchanged and focused on the female condition. In my opinion, this change is not acceptable, as it is not possible to title and discuss a paper focused on “couple infertility,” while the search criteria remained the previous ones, that is, referring to factors related to “female infertility.”

Rather, my critique was aimed at discussing the other part of the sentence, i.e., “determinants” of female infertility, either in the title or in the abstract and text. this statement suggests that male factors (particularly partners who abuse alcohol or chat) are determinants, or con-causal, or risk factors, for female infertility. This assumption is formally incorrect. They are associated factors, which means they are detected in women evaluated for infertility by questionnaires, and could certainly contribute to couple infertility, but not directly generate/cause female infertility. In fact, cross-sectional or retrospective studies cannot determine causal associations.

In summary, it remains a paper of potential interest. However, all the changes made to rename “female” infertility to “couple” infertility should be reinstated (according to the research criteria). Title and findings must meet the methodological standard of the studies included as well as clinical rationale, i.e., “factors associated " with female infertility, which differs from "risk factors" or "determinants" or "causal factors".

Line 308 - The main risk factor for STD is having multiple partners! This aspect can’t be missed in the discussion. The two conditions are most likely related.

“Maternal age” – Even though I respect the authors attitude to be adherent to previous publications, I would have made the same comments to those papers. Additionally, one of the publications included in the review specially refers to primary infertility. How can we name as “mother” someone who has never had the chance to conceive? Couldn't someone potentially feel offended?

Heterogeneity - Isn't the moderate to high heterogeneity of the included studies worth discussing?

Line 228-229, 374-375 – Please add hormonal factors to this list, at least in general, (i.e. we all know thyroid disorders, premature ovarian failure, etc.) as well as appropriate citations from international population-based reviews for this list. And what about cigarette smoking?

Line 352-354 - Please add appropriate references

Line 377-379 – The need of additional investigations about those factors just mentioned above in the Ethiopian population, as well as the need to invest significant resources to understand the impact of these conditions on infertility as a public health issue are among the main implications of the findings. Therefore, in my opinion they should be included in that paragraph more than just after limitations, as well as in conclusions and abstract!

Limitations – Was the separation between primary and secondary infertility not feasible for this study? This is a potential limitation as well as a relevant additional clinical implication of the findings. Additionally, an emphasis on this topic in the conclusions would also be worthwhile, helping as mentioned above to dictate future research directions essential to improve the understanding of this important public health issue.

7. PLOS authors have the option to publish the peer review history of their article (what does this mean? ). If published, this will include your full peer review and any attached files.

**Do you want your identity to be public for this peer review?** For information about this choice, including consent withdrawal, please see our Privacy Policy .

Reviewer #4: **Yes: ** Pietro Serra

Reviewer #5: **Yes: ** Michele Orsi

---

## [Author Response · Author response to Decision Letter 3]

9 Mar 2025

Authors’ Point-by-Point Response to the reviewers’ and editors’ Reports

Title: Factors associated with female infertility in Ethiopia: a systematic review and meta-analysis

Journal: PLOS ONE

Article type: Systematic review and Meta-analysis

Submission ID: PONE-D-24-14492R3

Point by point response to Reviewer 4’s comments

Comment: I carefully evaluated the revised version of this manuscript. Authors have performed the required changes, improving significantly the quality of the paper.

Response: Dear Dr. Pietro Serra, we sincerely appreciate your detailed and constructive feedback, which has greatly enhanced the quality of our manuscript. Your kind words, time, and effort in improving this paper mean a lot to us. Thank you again for your positive feedbacks.

Point by point response to Reviewer 5’s comments

Dear Dr. Michele Orsi,

We would like to acknowledge you for detailed and constructive comments, which helped us to improve the quality of the manuscript. We really appreciate your insightful critique, which has significantly strengthened the accuracy, clarity, and methodological rigor of our work. Thank you again for guiding us toward a more precise and scientifically sound presentation of our findings. Blow is our point-by-point response to each respective comment.

Comment: I think the authors have made a commendable effort to improve their manuscript. However, some aspects require further revision to meet the standards of this Journal. Thanks for adding line numbering.

Response: We are grateful for the reviewer’s thoughtful feedback and acknowledgment of our efforts to improve the manuscript. We have carefully addressed the remaining concerns to ensure the manuscript meets the journal’s standards. Please find our detailed responses to each comment below.

Comment: Line 1, 27, 37, 42, 84, 86, 120, 122, 226, 283, 290 etc. - I apologize if my previous comment was not entirely clear when I stated: “In my opinion, male factors including addiction to toxic substances abuse can't be formally considered among factors generating (or determinants of) female infertility. Although they are potential contributors to couple infertility, the direct impact on female health and reproductive function has yet to be demonstrated. The title and/or the implant of the review should be elaborated accordingly.” The criticism was not intended for the purpose of changing “female infertility” to “couple infertility,” while leaving the research criteria unchanged and focused on the female condition. In my opinion, this change is not acceptable, as it is not possible to title and discuss a paper focused on “couple infertility,” while the search criteria remained the previous ones, that is, referring to factors related to “female infertility.”

Rather, my critique was aimed at discussing the other part of the sentence, i.e., “determinants” of female infertility, either in the title or in the abstract and text. this statement suggests that male factors (particularly partners who abuse alcohol or chat) are determinants, or con-causal, or risk factors, for female infertility. This assumption is formally incorrect. They are associated factors, which means they are detected in women evaluated for infertility by questionnaires, and could certainly contribute to couple infertility, but not directly generate/cause female infertility. In fact, cross-sectional or retrospective studies cannot determine causal associations.

In summary, it remains a paper of potential interest. However, all the changes made to rename “female” infertility to “couple” infertility should be reinstated (according to the research criteria). Title and findings must meet the methodological standard of the studies included as well as clinical rationale, i.e., “factors associated with female infertility, which differs from "risk factors" or "determinants" or "causal factors".

Response: Thank you for your valuable and detailed comment. We sincerely appreciate your clarification regarding the distinction between "determinants" and "associated factors" in the context of female infertility. Initially, we did not place sufficient emphasis on the term "determinants" and used it in the title and text without fully considering its implications. We now understand that the term "determinants" implies a causal relationship, which is not supported by the cross-sectional nature of the primary studies included in our analysis.

We fully agree with your observation that male factors, such as addiction to toxic substances (e.g., alcohol or khat), cannot be formally considered determinants or causal factors of female infertility. While these factors may contribute to couple infertility, their direct impact on female reproductive health and function has not been demonstrated. Instead, these factors should be categorized as associated factors, as they were identified through questionnaires in cross sectional studies and may correlate with infertility but do not establish causation.

In line with your feedback, we have revised the manuscript to replace "determinants" with "factors associated with female infertility in Ethiopia." This adjustment ensures that the terminology aligns with the methodological standards of the included studies and the clinical rationale. Importantly, this change did not affect the search criteria, research framework, eligibility criteria, the total number of included and excluded studies, methodology and discussion of the study. We have also reinstated the focus on "female infertility" throughout the manuscript, as the search criteria and research framework were specifically designed to address factors associated with female infertility.

Comment: Line 308 - The main risk factor for STD is having multiple partners! This aspect can’t be missed in the discussion. The two conditions are most likely related.

Response: We sincerely appreciate your valuable feedback regarding the relationship between having multiple sexual partners and the risk of sexually transmitted disease (STDs). We would like to clarify that the association between having multiple sexual partners and female infertility was already discussed in Lines 304–314, and the association between multiple sexual partners and STDs was addressed in Lines 308–313. Additionally, the association between STDs and female infertility was thoroughly discussed in Lines 306–332. These sections collectively highlight the interconnected nature of these risk factors and their impact on female infertility. We believe these discussions adequately address the relationship between multiple sexual partners, STDs, and infertility as raised in your comment.

Comment: “Maternal age” – Even though I respect the authors attitude to be adherent to previous publications, I would have made the same comments to those papers. Additionally, one of the publications included in the review specially refers to primary infertility. How can we name as “mother” someone who has never had the chance to conceive? Couldn't someone potentially feel offended?

Response: Thank you for your insightful comment. We understand and appreciate your concern regarding the term “maternal age,” particularly in the context of individuals who have not had the opportunity to conceive. Our intention in using this term was to maintain consistency with the primary studies we referenced. However, we acknowledge that this terminology may not be fully inclusive, especially for individuals experiencing primary infertility.

To address this, we carefully reviewed our manuscript and replaced “maternal age” with more inclusive alternatives, such as “women's age,” through out the document. We believe these terms more accurately reflect the broader population under discussion without assuming prior conception or motherhood. This adjustment not only enhances the inclusivity of our work but also ensures greater precision in describing our study cohort.

Comment: Heterogeneity - Isn't the moderate to high heterogeneity of the included studies worth discussing?

Response: We sincerely appreciate your valuable feedback regarding the heterogeneity observed in the included studies. In response to your comment, we have explicitly mentioned the presence of moderate to high heterogeneity under each associated factor in the forest plots and discussed its implications in the Discussion section. Additionally, we have acknowledged heterogeneity as a limitation in the Limitations section, highlighting its potential impact on the generalizability of the findings. These revisions provide a comprehensive discussion of heterogeneity and its relevance to the interpretation of our results.

Comment: Line 228-229, 374-375 – Please add hormonal factors to this list, at least in general, (i.e. we all know thyroid disorders, premature ovarian failure, etc.) as well as appropriate citations from international population-based reviews for this list. And what about cigarette smoking?

Response: We sincerely appreciate your valuable feedback. In response to your comment, we have revised the manuscript to explicitly include hormonal factors (e.g., thyroid disorders, premature ovarian failure, hyperprolactinemia, and polycystic ovary syndrome) and cigarette smoking in the list of associated factors that were not comprehensively assessed in the primary studies. These factors are now mentioned in Lines 228–229 and 374–375.

Comment: Line 352-354 - Please add appropriate references

Response: Thank you for your valuable feedback. In response to your comment, we have revised Lines 352–354 and added appropriate references to support the statement. The revised text now includes citations from relevant studies that address the decline in egg quality and its impact on fertility with advancing maternal age.

Comment: Line 377-379 – The need of additional investigations about those factors just mentioned above in the Ethiopian population, as well as the need to invest significant resources to understand the impact of these conditions on infertility as a public health issue are among the main implications of the findings. Therefore, in my opinion they should be included in that paragraph more than just after limitations, as well as in conclusions and abstract!

Response: We sincerely appreciate your valuable feedback. In response to your comment, we have incorporated the understudied factors (e.g., hormonal imbalances, lifestyle, and socio-economic determinants) and recommendations for future research into the implications section, abstract, and conclusion section of the manuscript, as per your suggestion. These revisions emphasize the need for additional investigations into these factors and highlight the importance of investing resources to address infertility as a public health issue in Ethiopia. We believe these changes strengthen the manuscript and align it more closely with the public health priorities of the Ethiopian population.

Comment: Limitations – Was the separation between primary and secondary infertility not feasible for this study? This is a potential limitation as well as a relevant additional clinical implication of the findings. Additionally, an emphasis on this topic in the conclusions would also be worthwhile, helping as mentioned above to dictate future research directions essential to improve the understanding of this important public health issue.

Response: We sincerely appreciate your valuable feedback regarding the separation between primary and secondary infertility. In response to your comment, we acknowledge that the separation between primary and secondary infertility was not feasible for this study due to the lack of data in the primary studies. However, we recognize this as an important limitation and have emphasized its relevance in the Limitations section. Additionally, we have highlighted the need for future research to distinguish between primary and secondary infertility to improve the understanding of this important public health issue and inform targeted interventions under conclusion section.

---

## [Decision Letter · Decision Letter 4]

31 Mar 2025

PONE-D-24-14492R4Factors associated with female infertility in Ethiopia: A Systematic Review and Meta-AnalysisPLOS ONE

Dear Dr. Addisu,

Thank you for submitting your manuscript to PLOS ONE. After careful consideration, we feel that it has merit but does not fully meet PLOS ONE’s publication criteria as it currently stands. Therefore, we invite you to submit a revised version of the manuscript that addresses the points raised during the review process. Please submit your revised manuscript by May 15 2025 11:59PM. If you will need more time than this to complete your revisions, please reply to this message or contact the journal office at plosone@plos.org . Please include the following items when submitting your revised manuscript:

We look forward to receiving your revised manuscript.

Kind regards,

Assoc. Prof. Phakkharawat Sittiprapaporn, Ph.D.

Academic Editor

PLOS ONE

Journal Requirements:

Additional Editor Comments:

Reviewers' comments:

Reviewer's Responses to Questions

**Comments to the Author**

1. If the authors have adequately addressed your comments raised in a previous round of review and you feel that this manuscript is now acceptable for publication, you may indicate that here to bypass the “Comments to the Author” section, enter your conflict of interest statement in the “Confidential to Editor” section, and submit your "Accept" recommendation.

Reviewer #5: (No Response)

2. Is the manuscript technically sound, and do the data support the conclusions?

Reviewer #5: Yes

3. Has the statistical analysis been performed appropriately and rigorously? 

Reviewer #5: Yes

4. Have the authors made all data underlying the findings in their manuscript fully available?

Reviewer #5: Yes

5. Is the manuscript presented in an intelligible fashion and written in standard English?

Reviewer #5: Yes

6. Review Comments to the Author

Reviewer #5: I thank the Editorial Office for this opportunity, and commend the authors for their effort in revising their manuscript.

The last two comments:

- I appreciate they embedded the need of exploring the impact of additional factors in the discussion, conclusion and abstract. However, to make the abstract and conclusions clearer and straightforward, removal of details may be considered “(e.g., thyroid disorders, premature ovarian failure, hyperprolactinemia, and polycystic ovary syndrome)”. The list is already detailed in my opinion. “Gynecological conditions” may be the only element to add to the list, but details as for endocrinological conditions may be added to the discussion paragraph (e.g. uterine fibroids, endometrial polyps, endometriosis and adenomyosis).

- In the discussion paragraph, where the authors highlight those additional factors that need to be explored in future literature, detailed references to justify that claim should be added. We are aware that the review is not focused on those factors in different populations. However, the discussion should be justified by the existing literature, and potentially enriched by findings in optimally-resourced settings. Examples are provided below, and should be followed for all the conditions mentioned according to previous studies, in both similar and different settings.

Thyroid disorders

Huang Y, Xie B, Li J, Hang F, Hu Q, Jin Y, Qin R, Yu J, Luo J, Liao M, Qin A. Prevalence of thyroid autoantibody positivity in women with infertility: a systematic review and meta-analysis. BMC Womens Health. 2024 Nov 27;24(1):630. doi: 10.1186/s12905-024-03473-6. PMID: 39604908; PMCID: PMC11600930.

Endometriosis and adenomyosis

Vercellini P, Viganò P, Bandini V, Buggio L, Berlanda N, Somigliana E. Association of endometriosis and adenomyosis with pregnancy and infertility. Fertil Steril. 2023 May;119(5):727-740. doi: 10.1016/j.fertnstert.2023.03.018. Epub 2023 Mar 21. PMID: 36948440.

Uterine fibroids

Somigliana E, Reschini M, Bonanni V, Busnelli A, Li Piani L, Vercellini P. Fibroids and natural fertility: a systematic review and meta-analysis. Reprod Biomed Online. 2021 Jul;43(1):100-110. doi: 10.1016/j.rbmo.2021.03.013. Epub 2021 Mar 23. PMID: 33903032.

PCOS

Palomba S. Is fertility reduced in ovulatory women with polycystic ovary syndrome? An opinion paper. Hum Reprod. 2021 Aug 18;36(9):2421-2428. doi: 10.1093/humrep/deab181. PMID: 34333641.

7. PLOS authors have the option to publish the peer review history of their article (what does this mean? ). If published, this will include your full peer review and any attached files.

**Do you want your identity to be public for this peer review?** For information about this choice, including consent withdrawal, please see our Privacy Policy .

Reviewer #5: **Yes: ** Michele Orsi

---

## [Author Response · Author response to Decision Letter 4]

31 Mar 2025

Authors’ Point-by-Point Response to the reviewers’ and editors’ Reports

Title: Factors associated with female infertility in Ethiopia: a systematic review and meta-analysis

Journal: PLOS ONE

Article type: Systematic review and Meta-analysis

Submission ID: PONE-D-24-14492R4

Point by point response to editor comments

Comment: Please review your reference list to ensure that it is complete and correct. If you have cited papers that have been retracted, please include the rationale for doing so in the manuscript text, or remove these references and replace them with relevant current references. Any changes to the reference list should be mentioned in the rebuttal letter that accompanies your revised manuscript. If you need to cite a retracted article, indicate the article’s retracted status in the References list and also include a citation and full reference for the retraction notice.

Response: Dear Assoc. Prof. Phakkharawat Sittiprapaporn and the PLOS ONE Editorial Team,

Thank you for your note regarding the reference list. We have carefully reviewed all cited references in our manuscript and confirm that they are complete, correct, and free of retracted papers. Should any further adjustments be required, we are happy to address them promptly.

Point by point response to Reviewer 5’s comments

Dear Dr. Michele Orsi,

We would like to acknowledge you for detailed and constructive comments, which helped us to improve the quality of the manuscript. We really appreciate your insightful critique, which has significantly strengthened the accuracy, clarity, and methodological rigor of our work. Thank you again for guiding us toward a more precise and scientifically sound presentation of our findings. Blow is our point-by-point response to each respective comment.

Comment: The last two comments:

- I appreciate they embedded the need of exploring the impact of additional factors in the discussion, conclusion and abstract. However, to make the abstract and conclusions clearer and straightforward, removal of details may be considered “(e.g., thyroid disorders, premature ovarian failure, hyperprolactinemia, and polycystic ovary syndrome)”. The list is already detailed in my opinion. “Gynecological conditions” may be the only element to add to the list, but details as for endocrinological conditions may be added to the discussion paragraph (e.g. uterine fibroids, endometrial polyps, endometriosis and adenomyosis).

- In the discussion paragraph, where the authors highlight those additional factors that need to be explored in future literature, detailed references to justify that claim should be added. We are aware that the review is not focused on those factors in different populations. However, the discussion should be justified by the existing literature, and potentially enriched by findings in optimally-resourced settings. Examples are provided below, and should be followed for all the conditions mentioned according to previous studies, in both similar and different settings.

Response: We sincerely appreciate the reviewer's thoughtful feedback, which has helped strengthen our manuscript. In response to the final comments, we have streamlined the abstract and conclusions by removing the detailed listing of specific conditions (e.g., thyroid disorders, premature ovarian failure) and replacing them with the broader terms "gynecological and endocrinological conditions" to improve clarity and readability. The specific examples mentioned by the reviewer, including uterine fibroids, endometrial polyps, endometriosis and adenomyosis, have been incorporated into the discussion section where they can be properly contextualized. Regarding the discussion of factors needing further exploration, we have carefully reviewed the literature and added relevant references from both optimally-resourced settings and comparable populations to better substantiate our claims. These citations provide appropriate justification for highlighting these areas as needing further research while maintaining our focus on the study's primary objectives. We believe these revisions have significantly improved the manuscript.

---

## [Decision Letter · Decision Letter 5]

4 Apr 2025

Factors associated with female infertility in Ethiopia: A Systematic Review and Meta-Analysis

PONE-D-24-14492R5

Dear Dr. Addisu,

We’re pleased to inform you that your manuscript has been judged scientifically suitable for publication and will be formally accepted for publication once it meets all outstanding technical requirements.

Kind regards,

Assoc. Prof. Phakkharawat Sittiprapaporn, Ph.D.

Academic Editor

PLOS ONE

Reviewers' comments:

Reviewer's Responses to Questions

**Comments to the Author**

1. If the authors have adequately addressed your comments raised in a previous round of review and you feel that this manuscript is now acceptable for publication, you may indicate that here to bypass the “Comments to the Author” section, enter your conflict of interest statement in the “Confidential to Editor” section, and submit your "Accept" recommendation.

Reviewer #5: All comments have been addressed

2. Is the manuscript technically sound, and do the data support the conclusions?

Reviewer #5: Yes

3. Has the statistical analysis been performed appropriately and rigorously? 

Reviewer #5: Yes

4. Have the authors made all data underlying the findings in their manuscript fully available?

Reviewer #5: Yes

5. Is the manuscript presented in an intelligible fashion and written in standard English?

Reviewer #5: Yes

6. Review Comments to the Author

Reviewer #5: The authors have successfully revised the manuscript and now it sounds better integrated with the existing literature.

7. PLOS authors have the option to publish the peer review history of their article (what does this mean? ). If published, this will include your full peer review and any attached files.

**Do you want your identity to be public for this peer review?** For information about this choice, including consent withdrawal, please see our Privacy Policy .

Reviewer #5: **Yes: ** Michele Orsi

---

## [Editor Report · Acceptance letter]

PONE-D-24-14492R5

PLOS ONE

Dear Dr. Addisu,

I'm pleased to inform you that your manuscript has been deemed suitable for publication in PLOS ONE. Congratulations! Your manuscript is now being handed over to our production team.

Kind regards,

on behalf of

Assoc. Prof. Dr. Phakkharawat Sittiprapaporn

Academic Editor

PLOS ONE